

# Hyperbolic string tadpole

**Atakan Hilmi Firat⋆**

Center for Theoretical Physics, Massachusetts Institute of Technology,
Cambridge, MA 02139, USA

⋆ firat@mit.edu

## Abstract

Hyperbolic geometry on the one-bordered torus is numerically uniformized using Liouville theory. This geometry is relevant for the hyperbolic string tadpole vertex describing the one-loop quantum corrections of closed string field theory. We argue that the Lamé equation, upon fixing its accessory parameter via Polyakov conjecture, provides the input for the characterization. The explicit expressions for the Weil-Petersson metric as well as the local coordinates and the associated vertex region for the tadpole vertex are given in terms of classical torus conformal blocks. The relevance of this vertex for vacuum shift computations in string theory is highlighted.



## Contents



# 1  Introduction

Closed string field theory (CSFT) is a second-quantized formalism for string theory (for reviews see [1–4]). Despite the ongoing activity at multiple fronts in recent years [5–25], an *explicit* description of string vertices amenable to practical calculations is still lacking and this prevents further developments in CSFT. In particular, the construction of CSFT solutions are obscured primarily by our poor geometric understanding of the nature of string vertices.

In bosonic CSFT, the string vertices $\mathcal{V}_{g,n}$ are subsets of moduli spaces of Riemann surfaces of genus $g$ and $n$ punctures $\mathcal{M}_{g,n}$ endowed with a choice of local coordinates around each puncture up to a phase that satisfies *the geometric Batalin-Vilkovisky (BV) equation* [1]. Historically, the minimal-area metrics [26–32] are used to specify $\mathcal{V}_{g,n}$. Even though the minimal-area vertices lead to many insights on CSFT in the past, the existence issue for the higher genus surfaces persists and there is no clear efficient and systematic way to obtain a description for them, except for genus 0.

On the contrary, the hyperbolic string vertices of Costello and Zwiebach [33] work at the quantum level and it appears to be more amenable to an explicit description, see the developments [34–40]. These vertices are constructed by considering *bordered* Riemann surfaces endowed with hyperbolic metric (that is, the metric with constant negative curvature $K = -1$) whose borders have the length $L = 2\pi\lambda$ and grafting semi-infinite flat cylinders of the same circumference at each border. The grafted cylinders naturally provide the local coordinates around each puncture and the vertex regions for the moduli integration are specified by restricting to surfaces whose systoles are equal to or greater than $L$.[1] It is shown that hyperbolic string vertices solve the geometric BV equation if $0 < L \leq 2\operatorname{arcsinh} 1 \equiv L^* = 2\pi\lambda^*$ using the collar lemma [41].

Hyperbolic string vertices are recently related to Liouville theory and classical conformal blocks by the author [40]. It is shown that their local coordinates and the associated vertex regions can be constructed in the spirit of conformal bootstrap. This connection is intriguing and may eventually provide an improved understanding for the geometric input of the hyperbolic CSFT. The goal of this paper is to further elaborate on this emerging method in the case of higher genus surfaces that has been only sketched in [40] and construct the local coordinates and vertex region for *the hyperbolic string tadpole vertex*, hyperbolic tadpole for short. The construction here heavily relies on the known expressions of the classical torus conformal blocks [42,43].

To summarize, we demonstrate that the solutions $\psi(z)$ to the *Lamé equation*

$$\partial^2 \psi(z) + \frac{1}{2}(\delta \cdot \wp(z,\tau) + c)\psi(z) = 0, \tag{1}$$

---

[1] Systole of a Riemann surface is defined as the length of the shortest non-contractible curve that is non-homotopic to a boundary component.

can be used to obtain the local coordinates for the hyperbolic tadpole in the vertex and the Feynman regions on the $z$-plane with the identification $z \sim z + 1 \sim z + \tau$. Here $\wp(z, \tau)$ is *the Weierstrass elliptic function* and $\delta = 1/2 + \lambda^2/2$, with $L = 2\pi\lambda$ is the circumference of the grafted cylinder. *The accessory parameter $c$ as a function of the moduli of the torus $\tau$ and the length of the border $L$ is fixed in terms of a (version of) on-shell Liouville action* (40) upon using a (version of) Polyakov conjecture (30). Its expression for the vertex and Feynman regions are given in (60) and (79) respectively. These regions are demarcated by finding the length of the systole, see figure 5. A Mathematica package for the local coordinates is available upon request.

As a byproduct of our motivation from CSFT, we effectively provide a numerical characterization of the hyperbolic geometry on the one-bordered torus. That is, we obtained the length of the simple closed geodesics, as well as the hyperbolic metric, as a function of the moduli and the length of the border. Furthermore, we also derived *the Weil-Petersson (WP) metric* on the moduli space of the one-bordered torus as a series expansion and calculated its associated volume. Similar work along these lines has been performed for the four-punctured sphere in [44, 45] and for the four-bordered sphere in [40] in the limit $L \to \infty$. Extending them to finite $L$ amounts to a trivial work.

The local coordinates for the vertex region $\mathcal{V}_{1,1}$ and the Feynman region $\mathcal{F}_{1,1}$ can be used to compute off-shell one-loop diagrams systematically in (super-)string theory, in particular ones that appear in vacuum shift calculations, from *first-principles*. These calculations have been addressed in the past either using CSFT-inspired arguments [46, 47] or at the semi-formal level [48], with a suggestion of using $SL(2, \mathbb{C})$ vertices to eventually make it explicit. However, we point out that there are serious drawbacks of using $SL(2, \mathbb{C})$ vertices for systematic calculations involving a vacuum shift *and* mass renormalization currently; the local coordinates at the one-loop is semi-explicit and it is not clear how to extend $SL(2, \mathbb{C})$ vertices to remaining Riemann surfaces, especially to tori with two punctures. Since hyperbolic vertices are specified by a geometric prescription from the get-go these issues never rise. As long as the classical conformal blocks for Riemann surfaces are available everything can be determined explicitly.

The rest of the paper is organized as follows. We begin by detailing the procedure for how to solve for the local coordinates of quantum hyperbolic string vertices in section 2 and then specialize to the hyperbolic tadpole vertex. We present the Lamé equation and the Polyakov conjecture that determines its accessory parameter. The material in this section is primarily from [43], but we provide a detailed summary in order to set our conventions and fit into the framework of [40]. In section 3, we describe the conformal bootstrap procedure to uniformize the hyperbolic geometry. We compare our results with the exact expressions for special situations and check the modular crossing equation numerically. Here, we also find the WP metric and calculate its associated volume. Finally, we derive the vertex region and the local coordinates for the hyperbolic tadpole vertex in the subsequent two sections. We conclude our paper in section 6.

In appendices A and B we provide details on the special functions used in this work and our derivation of the classical torus conformal blocks after [42]. In appendix C we give additional details on our numerical results. We derive the Polyakov conjecture for tori with $n$ hyperbolic singularities in appendix D.

## 2 The Polyakov conjecture for the hyperbolic tadpole

In this section we introduce the Lamé Equation: the Fuchsian equation relevant for the hyperbolic tadpole vertex and argue for the Polyakov conjecture for tori with a single hyperbolic

singularity.[2] We begin by making general remarks on the behavior of Fuchsian equations on higher genus surfaces and then immediately specialize to the case of hyperbolic tadpole. We use the conventions and formalism of [40] with ingredients taken from [43].

## 2.1 The Fuchsian equation for higher genus surfaces

The (holomorphic) Fuchsian equation is given by

$$\partial^2\psi + \frac{1}{2}T(z)\psi = 0\,. \tag{2}$$

It is possible to use this equation and its hyperbolic monodromy problem to construct the local coordinates of classical hyperbolic vertices, as shown in [37, 40]. The hyperbolic monodromy problem asks for the conditions on $T(z)$ such that the solutions $\psi(z)$ can realize real ($PSL(2,\mathbb{R})$) hyperbolic monodromy around the punctures, called *hyperbolic singularities*. This problem is originally considered in [49, 50].

In this paper, we consider genus $g$ Riemann surfaces with $n$ hyperbolic singularities. The key point for the extension to non-vanishing genus is that the Fuchsian equation (2) is invariant under the conformal transformation $z \to \widetilde{z}$ as long as the objects $\psi(z)$ and $T(z)$ transform as

$$\psi(z) = \left(\frac{\partial\widetilde{z}}{\partial z}\right)^{-1/2}\widetilde{\psi}(\widetilde{z})\,, \qquad T(z) = \left(\frac{\partial\widetilde{z}}{\partial z}\right)^2\widetilde{T}(\widetilde{z}) + \{\widetilde{z}, z\}\,, \tag{3}$$

with the Schwarzian derivative $\{\cdot, \cdot\}$ is given by

$$\{\widetilde{z}, z\} = \frac{\partial^3\widetilde{z}}{\partial\widetilde{z}} - \frac{3}{2}\left(\frac{\partial^2\widetilde{z}}{\partial\widetilde{z}}\right)^2\,, \tag{4}$$

as usual. The transformation property (3) allows to use (2) on any given patch $z$ on the surface by taking (3) as their transition functions. Then the ideas and proofs for the genus 0 surfaces in [40] translates to higher genus surfaces word-by-word. In particular the equation (2), together with its solutions that realize hyperbolic $PSL(2,\mathbb{R})$ monodromy around each puncture, can be used to construct the local coordinates of quantum hyperbolic vertices. We point out that $T(z)$ in this context is commonly referred as *complex projective structure* [51].

Demanding a hyperbolic real monodromy around each puncture $p_i \in \Sigma_{g,n}(L_i)$ with $i = 1, \cdots, n$ forces $T(z)$ to contain double poles of residue $\delta_i > 1/2$ at each puncture. We call $\delta_i$ *the classical weights* and parameterize them as

$$\delta_i = \frac{1}{2} + \frac{\lambda_i^2}{2} = \frac{1}{2} + \frac{1}{2}\left(\frac{L_i}{2\pi}\right)^2\,. \tag{5}$$

In order to see why such a pole structure is present in $T(z)$, take a local coordinate patch $z$ around the puncture $p_i$ on the surface $\Sigma_{g,n}(L_i)$ and place $p_i$ at $z = 0$. Then we see

$$\partial^2\psi + \frac{1}{2}\frac{\delta_i}{z^2}\psi(z) + \cdots = 0 \implies \psi^\pm(z) \sim z^{1/2 \pm i\lambda_i}(1 + \cdots)\,. \tag{6}$$

That is, there is a basis of solutions that produces a (diagonal) real hyperbolic monodromy as a consequence of taking $z \to e^{2\pi i}z$. This is akin to the situation for the classical hyperbolic vertices. Notice that we indicate the dependence of the surfaces and the moduli spaces to the

---

[2]We often use the words puncture, hyperbolic singularity, border (of length $L$) and the grafted flat cylinder (of circumference $L$) interchangeably throughout this work, as these eventually describe the same situation as far as the hyperbolic vertices are concerned. The usual sense of a puncture, that is a *parabolic singularity* or *hyperbolic cusp*, corresponds to the case $L = 0$. We make the distinction when it may possibly lead to confusion.

parameters $L_i \equiv 2\pi\lambda_i$ by parenthesis. These parameters are associated to the circumference of the grafted flat cylinders of string vertices [37].

However, there is one crucial difference between the classical and quantum vertices in terms of how the rest of $T(z)$ is parameterized. Recall that there were $n-3$ undetermined accessory parameters that appeared as residues of the simple poles at the position of the punctures for genus 0 surfaces [40]. Such a "global" representation for $T(z)$ is not available for higher genus surfaces. Nevertheless, there are $3g-3+n$ undetermined complex accessory parameters contained in $T(z)$, which can be argued by considering the pants decomposition of Riemann surfaces [41]. Recall that all hyperbolic surfaces admits pant decomposition upon specifying $3g-3+n$ disjoint simple closed geodesics. Their lengths and twists provide a *local* coordinate in the moduli space. In hyperbolic geometry these coordinates are real. However, from the perspective of $T(z)$ and its associated metric, this is not the case since the monodromies are elements of $PSL(2,\mathbb{C})$, not $PSL(2,\mathbb{R})$, generally. So we should rather think that there are additional $3g-3+n$ parameters (the "imaginary" parts of the lengths and twists) that has been set to 0 in order to obtain real monodromies. This immediately shows that there should be additional $3g-3+n$ complex parameter on top of the usual complex moduli that has to be specified in $T(z)$ for the hyperbolic geometry. These are the accessory parameters.

Another way to argue for the existence of $3g-3+n$ undetermined complex accessory parameters is as follows [52]. Assume we have two complex projective structures $G(z)$ and $G'(z)$. Their difference $G(z)-G'(z)$ is a quadratic differential since it transforms as such by (3). Moreover, the vector space of quadratic differentials on an $n$ punctured genus $g$ surface $\Sigma$, $\mathcal{Q}(\Sigma)$, is $3g-3+n$ complex dimensional [53]. As a consequence, a generic complex projective structure $T(z)$ on $\Sigma$ can be parameterized as

$$T(z) = R(z) + \sum_{n=0}^{3g-3+n} \gamma_i Q_i(z), \tag{7}$$

where $R(z)$ is a reference complex projective structure and $\{Q_1(z), \cdots, Q_{3g-3+n}(z)\}$ is a basis for $\mathcal{Q}(\Sigma)$. Here $\gamma_i \in \mathbb{C}$ are the undetermined $3g-3+n$ complex accessory parameters of this parametrization that have to be fixed according to our demands on the monodromy. They depend on the choice of the reference $R(z)$ and the basis for $\mathcal{Q}(\Sigma)$. There exists ways to construct $R(z)$ directly given the surface $\Sigma$, for example, via symmetric bidifferential on $\Sigma$ [54]. A yet another argument for the existence of the accessory parameters for higher genus can be found in [55].

In the next subsection we specialize to the simplest quantum vertex $g = n = 1$ which contains a single complex accessory parameter. We are going to comment on the ways to make progress for remaining quantum vertices in conclusion 6 and appendix D.

## 2.2 The Lamé equation

In this subsection we specialize to tori with one hyperbolic singularity $\Sigma_{1,1}(L)$. They can be viewed as a quotient of the complex $z$-plane sans origin $\mathbb{C}^\times$ by a lattice $\Lambda = \mathbb{Z} + \tau\mathbb{Z}$ for $\tau = \tau_1 + i\tau_2 \in \mathbb{C}$,

$$\Sigma_{1,1}(L) \simeq \mathbb{C}^\times / \Lambda, \tag{8}$$

where the singularity is placed at the origin using the translation invariance. In other words, we identify $z \sim z + 1 \sim z + \tau$ for $z \in \mathbb{C}^\times$. It is sufficient to take $\tau \in \mathbb{H}$ as $\Lambda$ contains negative integer lattice points. In fact, $\tau$ belongs to the moduli space $\mathcal{M}_{1,1}(L)$ which is simply the quotient of the upper-half plane $\mathbb{H}$ by the modular group $PSL(2,\mathbb{Z})$

$$\tau \in \mathcal{M}_{1,1}(L) = \mathbb{H}/PSL(2,\mathbb{Z}) = \left\{ \tau \in \mathbb{H} \,\middle|\, |\mathrm{Re}\,\tau| \leq 1/2, \, |\tau| \geq 1, \, \tau \sim \tau + 1, \, \tau \sim -1/\tau \right\}, \tag{9}$$

as these describe inequivalent tori with a hyperbolic singularity. We call the set (9), considered as a subset of $\mathbb{H}$, *the fundamental domain*. Per usual, the action of the modular group $PSL(2,\mathbb{Z})$ is generated by the transformations

$$T: z \to z, \quad \tau \to \tau + 1, \qquad S: z \to \frac{z}{\tau}, \quad \tau \to -\frac{1}{\tau}. \tag{10}$$

For the hyperbolic tadpole, the only moduli is $\tau$ when we fix $L = 2\pi\lambda$ and the position of the singularity. Rather than working with $\Sigma_{1,1}(L)$ directly, we are going to work on $\mathbb{C}^\times$ after (8). This forces us to puncture the origin, as well as its images under the action of the lattice $\Lambda$, which makes the geometric quantities doubly-periodic. The coordinate $z$ refers to the global coordinate of $\mathbb{C}^\times$ henceforth unless stated otherwise.

Since having a hyperbolic monodromy demands having a double poles at the singularities with classical weights $\delta > 1/2$ as explained earlier in (6), $T(z)$ in (2) takes the form of

$$T(z) = \frac{\delta}{z^2} + \sum_{\lambda \in \Lambda \backslash \{0\}} \left[ \frac{\delta}{(z-\lambda)^2} - \frac{\delta}{\lambda^2} \right] + c = \delta \cdot \wp(z, \tau) + c, \tag{11}$$

by double periodicity. Notice that we have included a double pole (with appropriate subtraction factor for the convergence) for $z = 0$ and each of its images. The constant $c = c(\tau, \overline{\tau})$ is the single accessory parameter that should be fixed upon demanding a real hyperbolic monodromy around the puncture and its images. Crucially, simple poles at $z = 0$ and its images is absent because of the $z \to -z$ symmetry and regular terms are forbidden by the periodicity condition under the action of $\Lambda$. Thus we are naturally lead to consider the Weierstrass elliptic function $\wp(z, \tau)$. Some of its useful properties are listed in appendix A.

The relevant (holomorphic) Fuchsian equation for the hyperbolic tadpole is then given by

$$\partial^2 \psi(z) + \frac{1}{2}(\delta \cdot \wp(z, \tau) + c)\psi(z) = 0. \tag{12}$$

This is known as the Lamé equation whose solutions are *the Lamé functions* [43]. Before we solve this equation in order to find the local coordinates, the accessory parameter $c$ as a function of the moduli $\tau$ has to be found so that the solutions can realize a real hyperbolic monodromy around the puncture. This is equivalent to demanding $T(z)$ is given by

$$T(z) = -\frac{1}{2}(\partial \varphi)^2 + \partial^2 \varphi, \tag{13}$$

where $ds^2 = e^\varphi |dz|^2$ is the hyperbolic metric on the torus with a geodesic border of length $L \equiv 2\pi\lambda$. We observe that the Lamé equation (12) stays invariant under $z \to z + 1$ and $z \to z + \tau$ by (A.2), so it is indeed doubly periodic. Also we demand that the accessory parameter $c$ under the action of the modular group $PSL(2,\mathbb{Z})$ (10) changes as

$$T: c(\tau, \overline{\tau}) \to c(\tau + 1, \overline{\tau} + 1) = c(\tau, \overline{\tau}), \qquad S: c(\tau, \overline{\tau}) \to c\left(-\frac{1}{\tau}, -\frac{1}{\overline{\tau}}\right) = \tau^2 c(\tau, \overline{\tau}), \tag{14}$$

so that the Lamé equation (12) stays modular invariant. We have used the property (A.2) here and taken $\psi(z)$ to be invariant under the modular group. The anti-holomorphic counterpart is similar. As a consequence, the local coordinates would be modular-invariant as well.

Finally we point out the accessory parameter is endowed with an involution symmetry. That is

$$\tau \to -\overline{\tau} \implies c \to \overline{c}. \tag{15}$$

This can be argued taking the complex conjugate of (12) and noticing the torus with the moduli $\overline{\tau}$ is equivalent to the torus with the moduli $\tau = -\overline{\tau}$. The involution symmetry forces $c \in \mathbb{R}$ when $\text{Re}\,\tau = 0$. Combined with the constraints from the modular transformations (14) the involution symmetry further demands $\arg c = -\arg \tau$ for $|\tau| = 1$. This shows for $\tau = i, \exp(i\pi/3)$ we have $c = 0$ regardless of the value of the length of the border $2\pi\lambda$.

## 2.3 The Polyakov conjecture

In this subsection we determine the accessory parameter $c = c(\tau, \overline{\tau})$ of the Lamé equation (12) by considering the first non-trivial null state of the Virasoro algebra on the one-bordered torus, which we subsequently use to argue for the Polyakov conjecture for the hyperbolic tadpole. We are going to use the (modified) Liouville theory of [40]. The ideas here made an appearance in [43] before, but in the case of parabolic/elliptic singularities—we extend them to the hyperbolic singularities trivially. Like in [40], the methods here stem from heuristic path integral arguments so they don't consist of rigorous proofs. Nonetheless, we are going to justify the results by its consequences in the upcoming sections.

We begin with the relevant correlator for us, which is

$$\langle \Sigma_{1,1} \rangle_\tau \equiv \langle \mathcal{H}_\lambda(0,0) \rangle_\tau. \tag{16}$$

We indicated the dependence of the correlator on the moduli of the torus by the subscript $\tau$ and set the position of the "hole operator" (see [40]) to the origin using the translational symmetry. Here we have

$$\Delta = \frac{Q^2}{2}\delta = \frac{Q^2}{2}\left[\frac{1}{2} + \frac{\lambda^2}{2}\right] = \beta(Q - \beta), \qquad \beta = \frac{1}{2} + \frac{i\lambda}{2}. \tag{17}$$

As usual, $Q = b + b^{-1}$ and it is related to the central charge of Liouville's theory by $c = 1 + 6Q^2$. An important thing to notice that under the modular group the correlator (16) changes as

$$T : \langle \Sigma_{1,1} \rangle_\tau \to \langle \Sigma_{1,1} \rangle_{\tau+1} = \langle \Sigma_{1,1} \rangle_\tau, \qquad S : \langle \Sigma_{1,1} \rangle_\tau \to \langle \Sigma_{1,1} \rangle_{-\frac{1}{\tau}} = |\tau|^{2\Delta}\langle \Sigma_{1,1} \rangle_\tau, \tag{18}$$

using the weights (17) and the modular transformations (10).

We are interested in the first non-trivial null state of a Verma module. This is given by

$$|\chi_\pm\rangle = \left[L_{-2} - \frac{3}{2(2\Delta_\pm + 1)}L_{-1}\right]|\phi_\pm\rangle, \tag{19}$$

where $|\phi_\pm\rangle$ is a primary state of weight $\Delta_\pm$ where

$$\Delta_+ = -\frac{1}{2} - \frac{3b^2}{4}, \qquad \Delta_- = -\frac{1}{2} - \frac{3}{4b^2}, \tag{20}$$

and $L_{-n}$ are the Virasoro charges. We denote the fields associated to the states in (19) without a ket. Inserting the null field into the correlator in (16) leads to a decoupling equation

$$\left\langle \chi_+(z)\mathcal{H}_\lambda(\xi,\overline{\xi})\right\rangle_\tau = \left\langle \mathcal{L}_{-2}\phi_+(z)\mathcal{H}_\lambda(\xi,\overline{\xi})\right\rangle_\tau + \frac{1}{b^2}\left\langle \mathcal{L}_{-1}^2\phi_+(z)\mathcal{H}_\lambda(\xi,\overline{\xi})\right\rangle_\tau = 0. \tag{21}$$

Here $\mathcal{L}_{-n}$ are the Virasoro charges acting on the fields. Note that $\mathcal{L}_{-1} = \partial_z$ is the generator of translations. Also notice that we have

$$\left\langle \mathcal{L}_{-2}\phi_+(z)\mathcal{H}_\lambda(\xi,\overline{\xi})\right\rangle_\tau = \langle T_L(z)\phi_+(z)\mathcal{H}_\lambda(\xi,\overline{\xi})\rangle_\tau, \tag{22}$$

from the fact that $\mathcal{L}_{-2}\phi_+(z)$ appears in the correlator and it is equal to the constant term in $T_L\phi_+$ operator product expansion by the normal ordering. Here $T_L(z)$ is the stress-energy tensor of Liouville theory.

Now we use the conformal Ward identity on $n$-punctured tori derived in [56]. In particular, what we require are the equations (28) and (29), which we report here[3]

$$\langle T_L(z)X \rangle_\tau - \langle T_L(z) \rangle_\tau \langle X \rangle_\tau = \sum_{i=1}^n \Bigg[ \Delta_k \left( \wp(z - \xi_i, \tau) + 2\eta_1(\tau) \right)$$
$$+ \left( \zeta(z - \xi_i, \tau) + 2\eta_1(\tau)\xi_i \right) \partial_{\xi_i} \Bigg] \langle X \rangle_\tau + 2\pi i \partial_\tau \langle X \rangle_\tau, \tag{23}$$

---

[3]In our conventions $2\pi T^{there}(z) = T_L^{here}(z)$, see section 3 in [56] and [40].

and

$$\langle T_L(z)\rangle_\tau = 2\pi i \partial_\tau \log Z(\tau), \tag{24}$$

where $X = \phi_1(\xi_1)\cdots\phi_N(\xi_n)$ is a collection of primaries of weights $\Delta_i$ and $Z(\tau) \equiv \langle 1 \rangle_\tau$ is the partition function. We have used the following special functions in the expression above

$$\zeta(z,\tau) = \partial_z \log \vartheta_1(z|\tau) + 2\eta_1(\tau)z, \tag{25a}$$

$$\wp(z,\tau) = -\partial_z \zeta(z,\tau), \tag{25b}$$

$$\eta_1(\tau) = (2\pi)^2 \left[ \frac{1}{24} - \sum_{n=1}^\infty \frac{nq^n}{1-q^n} \right] = -2\pi i \partial_\tau \log \eta(\tau), \tag{25c}$$

where $\zeta(z,\tau)$ is the Weierstrass zeta function, $\vartheta_1(z|\tau)$ is the odd Jacobi theta function, and $\eta(\tau)$ is the Dedekind eta function whose conventions are given in appendix A. Observe that the Weierstrass elliptic function $\wp(z,\tau)$ has already introduced from different perspective in (11).

Taking $X = \phi_+(z)\mathcal{H}_\lambda(\xi,\overline{\xi})$ in (23) and subsequently using (22), (24) and (25), we see that the decoupling equation (21) takes the form of

$$\left[ \frac{1}{b^2}\partial_z^2 + (2\Delta_+\eta_1(\tau) + 2\eta_1(\tau)z\partial_z) + \Delta\left(\wp(z-\xi,\tau) + 2\eta_1(\tau)\right) \right. \tag{26}$$

$$\left. + \left(\zeta(z-\xi,\tau) + 2\eta_1(\tau)\xi\right)\partial_\xi + 2\pi i\partial_\tau \right]\left\langle \phi_+(z)\mathcal{H}_\lambda(\xi,\overline{\xi})\right\rangle_\tau + 2\pi i\partial_\tau \log Z(\tau) = 0.$$

We take $\xi \to 0$ using the translational symmetry on the torus in the subsequent analysis.

Now we are going to consider the semi-classical limit ($b \to 0$) of the equation (26). The important thing to notice here that $\phi_+$ field remains light and $\Delta_+ \to -1/2$, while the hole operator becomes heavy (i.e. it scales with $\sim 1/b^2$). So we expect that there will be a factorization

$$\langle \phi_+(z)\mathcal{H}_\lambda(0,0)\rangle_\tau \sim \phi_+^{cl}(z)\langle \Sigma_{1,1}\rangle_\tau, \tag{27}$$

where $\phi_+^{cl}(z)$ is the classical configuration for the field $\phi_+(z)$.

We can evaluate $\langle \Sigma_{1,1}\rangle_\tau$ using the saddle point approximation to the path integral

$$\langle \Sigma_{1,1}\rangle_\tau \sim \exp\left[ -\frac{1}{2b^2}S_{HJ}^{(1,1)}(\tau,\overline{\tau};\lambda) \right]. \tag{28}$$

Here $S_{HJ}^{(1,1)}(\tau,\overline{\tau};\lambda)$ is the on-shell action resulting from the (modified) Liouville theory on the torus. We call this *the on-shell Hadasz-Jaskólski (HJ) action* as in [40]. Note that it depends on the moduli $\tau$ and it's complex conjugate, as well as the parameter $\lambda$. It is a real function. We are going to discuss evaluation of this action in the next section.

Employing two equations above, we find the semi-classical limit of the equation (26) to be

$$\partial_z^2 \phi_+^{cl}(z) + \frac{1}{2}\left[ \delta\cdot\wp(z,\tau) + 2\delta\cdot\eta_1(\tau) - 2\pi i\partial_\tau S_{HJ}^{(1,1)}(\tau,\overline{\tau};\lambda) \right]\phi_+^{cl}(z) = 0. \tag{29}$$

This is nothing but the Lamé equation (12) whose accessory parameter is given

$$\boxed{c(\tau,\overline{\tau};\lambda) = 2\delta\cdot\eta_1(\tau) - 2\pi i\partial_\tau S_{HJ}^{(1,1)}(\tau,\overline{\tau};\lambda).} \tag{30}$$

This is the Polyakov conjecture for the torus with one hyperbolic singularity. We remark that the entire reasoning here can be generalized to the $n$-bordered torus by considering the decoupling equation (21) upon insertion of additional hole operators in the correlator. Since this is

outside of the main development of the paper and for completeness we present its derivation in appendix D.

Let us remark on the relation (30). First, the choice of the accessory parameter $c$ in (30) guarantees that the Lamé equation (12) can realize a real hyperbolic monodromy around the puncture and its images. The justification for this as follows. The second derivative term in (29) purely comes from the $\mathcal{L}_{-1}^2$ term in (21), while the rest of the terms comes from $\mathcal{L}_{-2}$ which is related to the correlator $\langle T_L(z)\phi_+(z)\Sigma_{1,1}\rangle_\tau$ as explained in (22). This correlator factorizes in the semi-classical limit and $\langle\Sigma_{1,1}\rangle_\tau$ factor drops out of the equation (29) and we are left with

$$\langle T_L(z)\rangle \sim T_L^{cl}(z), \quad \text{where} \quad T_L^{cl}(z) = \frac{1}{2b^2}\left(\delta\cdot\wp(z,\tau) + 2\,\delta\cdot\eta_1(\tau) - 2\pi i\partial_\tau S_{HJ}^{(1,1)}(\tau,\overline{\tau};\lambda)\right). \tag{31}$$

The expression inside the parenthesis is precisely the stress-energy tensor (13) associated with the hyperbolic metric [40]. Additionally, we see that $\phi_+^{cl}(z)$ is related to the weight $-1/2$ primaries $\psi(z)$ used to construct the local coordinates in (2).

Deriving the Polyakov conjecture from the decoupling equation (21) and interpreting the classical null field $\phi_+^{cl}(z)$ as a weight $-1/2$ primary is not special to the case here: it holds for any Riemann surface. In particular, we can run a similar argument for genus 0 surfaces, for details see [57]. This provides an alternative argument to the one used in [40]. For higher genus surfaces, on the other hand, the derivation by the decoupling equation sketched above is more accessible.

One of the important checks for the conjecture (30) is to test its consistency with the involution (15) and modular symmetries (10). The consistency for the involution symmetry is apparent given $S_{HJ}^{(1,1)}$ is a real function. The consistency for the modular symmetry can be established by noticing $S_{HJ}^{(1,1)}$ have the following modular transformations

$$T : S_{HJ}^{(1,1)}(\tau,\overline{\tau};\lambda) \to S_{HJ}^{(1,1)}(\tau+1,\overline{\tau}+1;\lambda) = S_{HJ}^{(1,1)}(\tau,\overline{\tau};\lambda), \tag{32a}$$

$$S : S_{HJ}^{(1,1)}(\tau,\overline{\tau};\lambda) \to S_{HJ}^{(1,1)}\left(-\frac{1}{\tau},-\frac{1}{\overline{\tau}};\lambda\right) = S_{HJ}^{(1,1)}(\tau,\overline{\tau};\lambda) - 2\delta\log|\tau|, \tag{32b}$$

by (18) and (28). This immediately shows the Polyakov conjecture (30) is consistent with the $T$ transformation by (A.10). It is also consistent with the $S$ transformation via (14), (25c) and (A.10).

Given the Polyakov conjecture for the hyperbolic tadpole (30), solving the hyperbolic monodromy problem turns into determining $S_{HJ}^{(1,1)}(\tau,\overline{\tau};\lambda)$, i.e. the on-shell action of the (modified) Liouville theory on the torus. It is possible to construct this action via classical modular conformal bootstrap. We evaluate $S_{HJ}^{(1,1)}(\tau,\overline{\tau};\lambda)$ as a function of the moduli $\tau$ and the length of the border $L = 2\pi\lambda$ in the next section.

## 3 Uniformizing one-bordered torus

In this section we uniformize the hyperbolic geometry on the one-bordered torus. By this we mean finding the length of the simple closed geodesic of one-bordered torus as a function of moduli and the length of the border, evaluating the on-shell HJ action $S_{HJ}^{(1,1)}(\tau,\overline{\tau})$ specifying the accessory parameter of (12) through (30) and solving *the Weil-Petersson (WP) metric* on the moduli space. We test our results by comparing them with the exact results at the symmetric points and checking modular crossing, as well as computing the WP volume of the moduli space $\mathcal{M}_{1,1}$.

We begin by considering the correlator (16). Like in [40], we use the operator formalism to write the following *modular* bootstrap equation

$$\langle \mathcal{H}_\lambda(0,0) \rangle_\tau = \int\limits_{\frac{Q}{2}(1+i\mathbb{R}^+)} d\lambda' \, \widetilde{C}(\lambda',\lambda,-\lambda') e^{Q^2\lambda'^2 s/2} |\mathcal{F}^\Delta_{1+6Q^2,\Delta'}(q)|^2 . \tag{33}$$

Let us describe the equality (33) in more detail. Here $\widetilde{C}$ is the reflection-symmetric DOZZ formula for the three-point function of Liouville theory. Only its semi-classical limit is relevant for us and this is evaluated in [50] for hyperbolic singularities. It is given by

$$\widetilde{C}(\lambda_3,\lambda_2,\lambda_1) \sim \exp\left[-\frac{Q^2}{2} S^{(0,3)}_{HJ}(\lambda_3,\lambda_2,\lambda_1)\right], \tag{34}$$

where $S^{(0,3)}_{HJ}(\lambda_3,\lambda_2,\lambda_1)$ is the on-shell HJ action on the sphere with three hyperbolic singularities, whose expression is given by

$$S^{(0,3)}_{HJ}(\lambda_3,\lambda_2,\lambda_1) = 2 \sum_{\sigma_2,\sigma_3=\pm} F\left(\frac{1}{2} + \frac{i\lambda_1}{2} + \sigma_2\frac{i\lambda_2}{2} + \sigma_3\frac{i\lambda_3}{2}\right) + 2 \sum_{j=1}^{3}\left[H(i\lambda_j) + \frac{\pi}{2}|\lambda_j|\right], \tag{35}$$

for $\lambda_i \in \mathbb{R}$ up to an irrelevant additive constant. The functions $F$ and $H$ are defined by

$$F(x) \equiv \int_{\frac{1}{2}}^{x} dy \log \frac{\Gamma(y)}{\Gamma(1-y)}, \qquad H(x) \equiv \int_{0}^{x} dy \log \frac{\Gamma(-y)}{\Gamma(y)}. \tag{36}$$

The on-shell action $S^{(0,3)}_{HJ}(\lambda_3,\lambda_2,\lambda_1)$ is invariant under flipping the sign of its arguments and totally symmetric by construction.

In (33), we have taken two of the arguments of $\widetilde{C}$ equal and opposite of each other and integrate over them. This is because we are supposed to identify two hole operators with border length $L = 2\pi\lambda'$ in the generalized hyperbolic three-vertex to construct the hyperbolic tadpole. We additionally included $e^{Q^2\lambda'^2 s/2}$ in order embed a possible finite flat cylinder of circumference $2\pi\lambda'$ in the geometry. The external hole operator has the associated border length $2\pi\lambda$. We take $\lambda, \lambda' \geq 0$ without loss of generality.

Finally, the function $\mathcal{F}^\Delta_{1+6Q^2,\Delta'}(q)$ in (33) is the torus conformal blocks and it is entirely determined by the Virasoro algebra as a function of the moduli $q = e^{2\pi i\tau}$. It depends on the central charge $c = 1+6Q^2$ and the conformal weights $\Delta, \Delta'$ of the external and internal operators respectively. Its semi-classical limit, *the classical torus conformal blocks $f^\lambda_{\lambda'}(q)$ is expected to be related to the torus conformal blocks by [43]

$$\mathcal{F}^\Delta_{1+6Q^2,\Delta'}(q) \overset{Q\to\infty}{\sim} \exp\left[Q^2 f^\lambda_{\lambda'}(q)\right]. \tag{37}$$

Although there is no rigorous proof of this relation, like in the case of the four-punctured sphere [58], the non-trivial exponentiation behavior is well-supported by the expansion of the torus conformal block, which can be found by a recursion [42]. More details on this recursion are given in appendix B.

We are interested in the semi-classic limit ($Q \to \infty$) of the expression (33), as this is expected to describe the hyperbolic geometry in question. Combining the remarks above, together with (28), we see that

$$\exp\left[-\frac{Q^2}{2} S^{(1,1)}_{HJ}(\tau,\overline{\tau};\lambda)\right] \sim \int_{0}^{\infty} d\lambda' \exp\left[-\frac{Q^2}{2}\left(S^{(0,3)}_{HJ}(\lambda',\lambda,-\lambda') - \lambda'^2 s - 2f^\lambda_{\lambda'}(q) - 2\overline{f}^\lambda_{\lambda'}(\overline{q})\right)\right]. \tag{38}$$

Here the bar indicates the complex conjugation. This integral is dominated by the saddle point at $\lambda' = \lambda_s(\tau, \overline{\tau}; \lambda)$ in the $Q \to \infty$ limit and the action $S_{HJ}^{(1,1)}(\tau, \overline{\tau}; \lambda)$ is evaluated by solving

$$\frac{\partial}{\partial \lambda'} \left[ S_{HJ}^{(0,3)}(\lambda', \lambda, -\lambda') - \lambda'^2 s - 2f_{\lambda'}^{\lambda}(q) - 2\overline{f}_{\lambda'}^{\lambda}(\overline{q}) \right]_{\lambda' = \lambda_s(\tau, \overline{\tau}; \lambda)} = 0 \,. \tag{39}$$

This produces

$$S_{HJ}^{(1,1)}(\tau, \overline{\tau}; \lambda) = S_{HJ}^{(0,3)}(\lambda_s, \lambda, -\lambda_s) - \lambda_s^2 s - 2f_{\lambda_s}^{\lambda}(q) - 2\overline{f}_{\lambda_s}^{\lambda}(\overline{q}) \,, \tag{40}$$

and the accessory parameter $c$ (30) reads,

$$c(q, \overline{q}; \lambda) = (1 + \lambda^2) \, \eta_1(q) + 4\pi^2 q \, \frac{\partial S_{HJ}^{(1,1)}(q, \overline{q}; \lambda_s)}{\partial q} = (1 + \lambda^2) \, \eta_1(q) - 8\pi^2 q \, \frac{\partial f_{\lambda'}^{\lambda}(q)}{\partial q} \bigg|_{\lambda' = \lambda_s(\tau, \overline{\tau}; \lambda)} \,. \tag{41}$$

## 3.1 The saddle-point and the length of the simple closed geodesic

As in [40], the semi-classical expectation suggests that the length of the simple non-contractible geodesic (called simply as *internal geodesic*) is $2\pi\lambda_s$ on $\Sigma_{1,1}(L)$ and the length of the string propagator to be proportional to $s$. We set $s = 0$ for now and come back to $s > 0$ case relevant for Feynman diagrams in the next section.

We begin by solving for the saddle-point (39). For this we need the derivative of $S_{HJ}^{(0,3)}(\lambda', \lambda, -\lambda')$ and $f_{\lambda'}^{\lambda}(q)$. Let us focus on the first one. As pointed out in [50], we have

$$\frac{\partial S_{HJ}^{(0,3)}}{\partial \lambda'}(\lambda', \lambda, -\lambda') = -2\pi + 2i \log \left[ \gamma \left( \frac{1}{2} + \frac{i\lambda}{2} + i\lambda' \right) \gamma \left( \frac{1}{2} - \frac{i\lambda}{2} + i\lambda' \right) \frac{\Gamma(1 - i\lambda')^2}{\Gamma(1 + i\lambda')^2} \right]$$
$$= 4\lambda' \log R(\lambda', \lambda, -\lambda') \,, \tag{42}$$

where $\gamma(z) \equiv \Gamma(z)/\Gamma(1-z)$. The extra $\pi$'s come from using the gamma function identity $\Gamma(z + 1) = z\,\Gamma(z)$ and their associated branch differences after integrating. We point out the right-hand side is related to the mapping radius of the (generalized) hyperbolic three-vertex [37]

$$R(\lambda_1, \lambda_2, \lambda_3) = e^{-\pi/2\lambda_1} \left[ \frac{\Gamma(1 - i\lambda_1)^2}{\Gamma(1 + i\lambda_1)^2} \frac{\gamma\left(\frac{1}{2}(1 + i\lambda_1 + i\lambda_2 + i\lambda_3)\right) \gamma\left(\frac{1}{2}(1 + i\lambda_1 - i\lambda_2 + i\lambda_3)\right)}{\gamma\left(\frac{1}{2}(1 - i\lambda_1 - i\lambda_2 + i\lambda_3)\right) \gamma\left(\frac{1}{2}(1 - i\lambda_1 + i\lambda_2 + i\lambda_3)\right)} \right]^{i/2\lambda_1} \,, \tag{43}$$

which we have used in (42).

It is advantageous to consider the series expansion of (42) in $\lambda'$. Fortunately, we can obtain it in closed form using the polygamma functions $\psi$, see (A.13) for conventions. This is simply due to

$$\log \Gamma \left( a + i\lambda' \right) = \log \Gamma(a) + \sum_{n=1}^{\infty} \left[ \frac{\psi^{(n-1)}(a)}{n!} \, (i\lambda')^n \right] \,. \tag{44}$$

After some algebra and using (A.14a) it can be shown that

$$\frac{\partial S_{HJ}^{(0,3)}}{\partial \lambda'}(\lambda', \lambda, -\lambda') = \sum_{n=0}^{\infty} s_n(\lambda)(\lambda')^n \,, \tag{45}$$

where the coefficients $s_n(\lambda)$ are given by

$$s_n(\lambda) = \begin{cases} -2\pi \,, & \text{if } n = 0, \\ -8 \left[ \gamma + \operatorname{Re} \psi^{(0)} \left( \frac{1}{2} + \frac{i\lambda}{2} \right) \right], & \text{if } n = 1, \\ 0 \,, & \text{if } n \in 2\mathbb{Z}_{\geq 1}, \\ \frac{8(-1)^{[n/2]}}{n} \left[ \frac{\operatorname{Re} \psi^{(n-1)}\left(\frac{1}{2} + \frac{i\lambda}{2}\right)}{(n-1)!} + \zeta(n) \right], & \text{if } n \in 2\mathbb{Z}_{\geq 1} + 1. \end{cases} \tag{46}$$

These coefficients can be further simplified for the punctured torus ($\lambda = 0$) using (A.14b).

We also need to take derivative of the torus conformal blocks with respect to $\lambda'$. This is

$$\frac{\partial f_{\lambda'}^\lambda(q)}{\partial \lambda'} = \frac{\lambda'}{2}\log q - \frac{(1+\lambda^2)^2\lambda'}{4(1+\lambda'^2)^2}q + \mathcal{O}(q^2) = \frac{\lambda'}{2}\log q - \left[\frac{1}{4}(1+\lambda^2)^2\lambda' + \cdots\right]q + \mathcal{O}(q^2). \quad (47)$$

We point that in the second line we have expanded in $\lambda'$ like we did for derivative of $S_{HJ}^{(0,3)}$ and dots stand for this expansion.

For a moment, let us focus on the $\log q$ part of the classical conformal block and ignore the remaining higher powers in $q$ and consider the linear term in $\lambda'$ in (45). The saddle-point equation (39) reads

$$0 = -2\pi - 8\lambda_s\left[\gamma + \mathrm{Re}\,\psi^{(0)}\left(\frac{1}{2} + \frac{i\lambda}{2}\right)\right] - \lambda_s \log|q|^2 + \cdots \quad (48)$$

$$\implies \lambda_s = \frac{-2\pi}{8\left[\gamma + \mathrm{Re}\,\psi^{(0)}\left(\frac{1}{2} + \frac{i\lambda}{2}\right)\right] + \log|q|^2} + \cdots = \frac{2\pi}{s_1(\lambda) - \log|q|^2} + \cdots$$

For convenience, we define the parameter

$$\xi \equiv \frac{2\pi}{s_1(\lambda) - \log|q|^2}. \quad (49)$$

Now it is possible to set up a recursive procedure to compute $\lambda_s$ as an expansion of $\xi$. For example, the saddle-point equation to order $\lambda_s^3$ is given by

$$0 = -2\pi + \lambda_s s_1(\lambda) + \lambda_s^3 s_3(\lambda) + \cdots - \lambda_s \log|q|^2 + \cdots \implies \lambda_s = \xi - \frac{\xi}{2\pi}s_3(\lambda)\lambda_s^3 + \cdots \quad (50)$$

Plugging this equation back into itself we find

$$\lambda_s = \xi - \frac{\xi^4}{2\pi}s_3(\lambda) + \cdots = \xi - \frac{2\xi^4}{3\pi}\left[\frac{\mathrm{Re}\,\psi^{(2)}\left(\frac{1}{2} + \frac{i\lambda}{2}\right)}{2} + 2\zeta(3)\right]\xi^4 + \cdots \quad (51)$$

This procedure can be repeated to arbitrarily high orders to find $\lambda_s$ as an expansion of $\xi$.

Before we do that, let us discuss the inclusion of higher powers of $q$ to this expansion. Keeping only the linear terms in $\lambda'$ again, the saddle-point equation (39) takes the form

$$0 = -2\pi + \lambda_s s_1(\lambda) - \lambda_s \log|q|^2 + \lambda_s(1+\lambda^2)^2\,\mathrm{Re}(q) + \cdots \quad (52)$$

$$\implies \lambda_s = \frac{2\pi}{s_1(\lambda) - \log|q|^2 + (1+\lambda^2)^2\,\mathrm{Re}(q) + \cdots} + \cdots = \xi - \frac{\xi^2}{2\pi}(1+\lambda^2)^2\,\mathrm{Re}(q) + \cdots$$

We can also write a recursion similar to (50) and then use the geometric series to expand the denominator. In the language of [44, 45], the expansion in $q$ can be understood as a "non-perturbative" correction on top of the $\xi$ series because we have $q \sim e^{-1/\xi^2}$. We point that these corrections are highly suppressed relative to the "perturbative" series for $|\tau| > 1$.

In summary, we conclude that $\lambda_s$ can be found as a double expansion in $\xi$ and $q$:

$$\lambda_s = \xi + \left[-\frac{(1+\lambda^2)^2}{2\pi}\mathrm{Re}(q) + \frac{3(1+\lambda^2)^2(-75 + 154\lambda^2 + 37\lambda^4)}{1024\pi}\mathrm{Re}(q^2) + \cdots\right]\xi^2$$

$$+ \left[\frac{(1+\lambda^2)^4}{4\pi^2}\mathrm{Re}(q)^2 + \cdots\right]\xi^3 \quad (53)$$

$$+ \left[-\frac{2\mathrm{Re}\,\psi^{(2)}\left(\frac{1}{2} + \frac{i\lambda}{2}\right) + 4\zeta(3)}{3\pi} + \frac{(1+\lambda^2)^2}{\pi}\mathrm{Re}(q)\right.$$

$$\left. - \frac{3(1+\lambda^2)^2(-27 + 1018\lambda^2 + 341\lambda^4)}{2048\pi}\mathrm{Re}(q^2) + \cdots\right]\xi^4 + \cdots$$

Table 1: The comparison between the exact and computed results for the length of the internal geodesics for the square ($\tau = i$) and rhombus ($\tau = e^{\pi i/3}$) torus with a border of length 0 and $L^* = 2 \operatorname{arcsinh} 1$. The exact results are taken from [59].

| Moduli $\tau$ | Border length $L$ | Exact result | Computed result | Relative error |
|---|---|---|---|---|
| $i$ | 0 | $2 \operatorname{arccosh} \sqrt{2} \approx 1.7627471740$ | 1.7627471745 | $4.03 \times 10^{-10}$ |
| $e^{i\pi/3}$ | 0 | $2 \operatorname{arccosh} 3/2 \approx 1.9248473$ | 1.9248475 | $1.20 \times 10^{-7}$ |
| $e^{i\pi/3}$ | $2 \operatorname{arcsinh} 1 \approx 1.76$ | $\approx 2.0006589936$ | 2.0006589940 | $1.72 \times 10^{-10}$ |

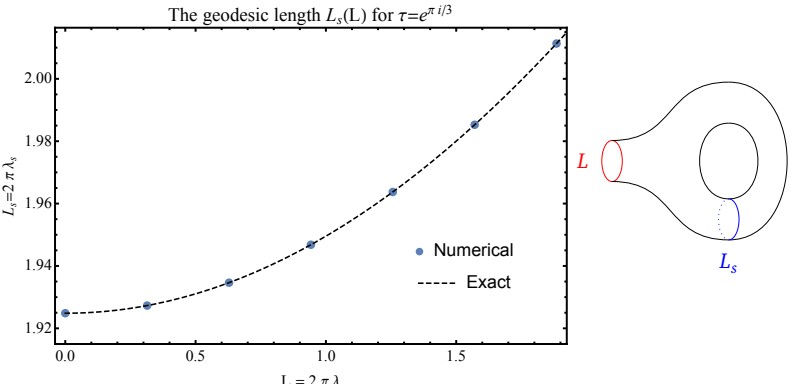

Figure 1: The progression of the length of the internal geodesic $L_s = 2\pi\lambda_s$ as a function of the length of the border $L = 2\pi\lambda$ for the rhombus torus $\tau = e^{i\pi/3}$. The black dashed line is the exact result obtained from [59].

We have obtained the expansion to the order $\mathcal{O}(\xi^{19}, q^4)$. Setting $\lambda$ to a precise value allows us to go even higher-orders in the expansion so we always used the highest order possible for our result. In the context of hyperbolic CSFT, the most natural value for $\lambda$ is $\lambda = \lambda^* = \operatorname{arcsinh} 1/\pi$ [33], so we often investigate this case explicitly. However, we additionally investigate $\lambda = 0$ (i.e. one-punctured torus) because of its simplicity. We point out this procedure is similar to [44, 45] and different from the one in [40]. In principle, it is possible to do something similar for the case considered in [40].

The comparison of our results with the exact results by Maskit [59] is shown in table 1 and figure 1. The agreement is stellar and the relative errors are usually less than $\sim 10^{-7}$. Further evidence of the convergence of the series (53) is given in appendix C.

## 3.2 The on-shell HJ action, modular symmetry, and the accessory parameter

Having a saddle-point at $\lambda' = \lambda_s$ allows us to write down the on-shell action $S_{HJ}^{(1,1)}(\tau, \overline{\tau}; \lambda)$ as an expansion in $\xi$ and $q$ via (40). A crucial observation here is that $S_{HJ}^{(0,3)}$ has the following expansion in $\lambda'$ by integrating (45)

$$S_{HJ}^{(0,3)}(\lambda', \lambda, -\lambda') = s_{-1}(\lambda) + \sum_{n=1}^{\infty} \frac{s_{n-1}(\lambda)}{n} (\lambda')^n. \tag{54}$$

Here $s_{-1}(\lambda)$ is given by evaluating $S_{HJ}^{(0,3)}(0, \lambda, 0)$

$$s_{-1}(\lambda) = S_{HJ}^{(0,3)}(0, \lambda, 0) = 8F\left(\frac{1}{2} + \frac{i\lambda}{2}\right) + 2H(i\lambda) + \pi\lambda, \tag{55}$$

where we have used the fact $F(1/2+x) = F(1/2-x)$, see (36). Take note that $s_{-1}(\lambda = 0) = 0$. Similarly, we will expand the classical conformal block (B.4) in $\lambda'$.

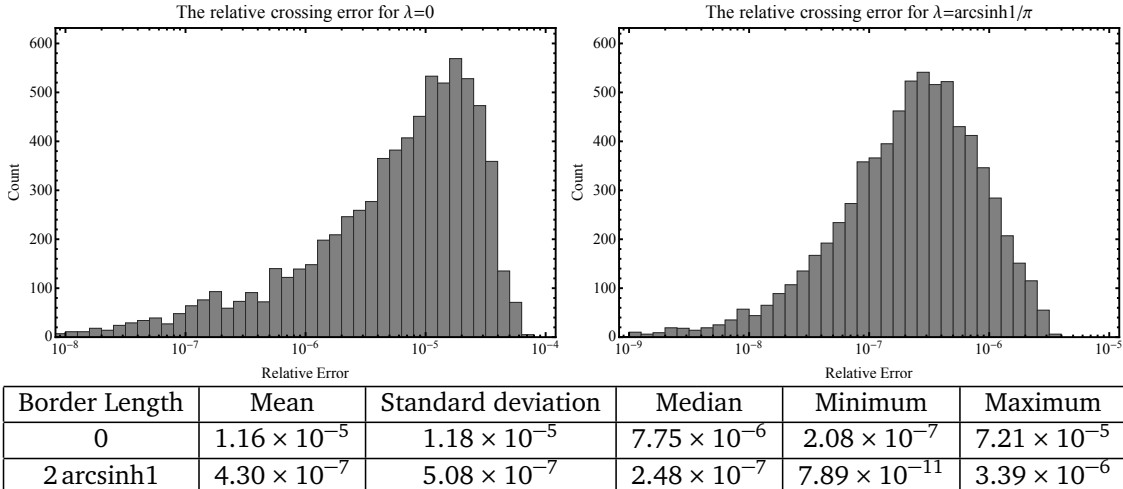

| Border Length | Mean | Standard deviation | Median | Minimum | Maximum |
|---|---|---|---|---|---|
| 0 | $1.16 \times 10^{-5}$ | $1.18 \times 10^{-5}$ | $7.75 \times 10^{-6}$ | $2.08 \times 10^{-7}$ | $7.21 \times 10^{-5}$ |
| $2\operatorname{arcsinh}1$ | $4.30 \times 10^{-7}$ | $5.08 \times 10^{-7}$ | $2.48 \times 10^{-7}$ | $7.89 \times 10^{-11}$ | $3.39 \times 10^{-6}$ |

Figure 2: The distribution of the relative errors for the crossing equation (58) for $\lambda = 0$ and $\lambda = \operatorname{arcsinh}1/\pi$. The points are sampled from the fundamental domain with $\mathrm{Im}(\tau) < 1.2$. We observed errors tend to increase for larger values of $\mathrm{Im}(\tau)$.

The on-shell action $S_{HJ}^{(1,1)}(\tau, \overline{\tau}; \lambda)$ is then given by

$$S_{HJ}^{(1,1)}(\tau, \overline{\tau}; \lambda) = s_{-1}(\lambda) + s_0(\lambda)\lambda_s + \frac{s_1(\lambda)}{2}\lambda_s^2 + \cdots \tag{56}$$

$$- \frac{\lambda_s^2}{2}\log|q|^2 - \left[\frac{(1+\lambda^2)^2}{2} - \frac{(1+\lambda^2)^2}{2}\lambda_s^2 + \cdots\right]\mathrm{Re}(q) + \cdots$$

$$= 8F\left(\frac{1}{2} + \frac{i\lambda}{2}\right) + 2H(i\lambda) + \pi|\lambda| - 2\pi\lambda_s + \frac{\pi}{\xi}\lambda_s^2 + \cdots$$

$$- \left[\frac{(1+\lambda^2)^2}{2} - \frac{(1+\lambda^2)^2}{2}\lambda_s^2 + \cdots\right]\mathrm{Re}(q) + \cdots,$$

where we have combined the leading term coming from classical torus conformal blocks with the $\lambda_s^2$ term in $S_{HJ}^{(0,3)}$. Since we have $\lambda_s \sim \xi$, we can plug (53) in the expansion above and that would produce a double expansion in $\xi$ and $q$ given by

$$S_{HJ}^{(1,1)}(\tau, \overline{\tau}; \lambda) = \left[8F\left(\frac{1}{2} + \frac{i\lambda}{2}\right) + 2H(i\lambda) + \pi|\lambda| - \frac{(1+\lambda^2)^2}{2}\mathrm{Re}(q) + \cdots\right] - \pi\xi$$

$$+ \left[\frac{1}{2}(1+\lambda^2)^2\,\mathrm{Re}(q) + \cdots\right]\xi^2 + \cdots \tag{57}$$

Note that $\mathcal{O}(\xi)$ term won't receive any non-perturbative correction because of the construction in (56) and due to $\lambda_s \sim \xi$ at the leading order in $\xi$. We further observe the coefficient of $\mathcal{O}(\xi^0)$ terms always contain terms of the form $\mathrm{Re}(q^n)$, but this will not be the case for higher orders.

The on-shell action $S_{HJ}^{(1,1)}(\tau, \overline{\tau}; \lambda)$ has to satisfy the modular crossing

$$S_{HJ}^{(1,1)}(\tau, \overline{\tau}; \lambda) = S_{HJ}^{(1,1)}\left(-\frac{1}{\tau}, -\frac{1}{\overline{\tau}}; \lambda\right) + (1+\lambda^2)\log|\tau|, \tag{58}$$

as a result of its modular invariance property given in (32). In order to test its validity we randomly sampled $\approx 7.5 \times 10^3$ points in $\mathcal{M}_{1,1}(L)$, evaluated both sides of (58), and calculated their relative errors. The results are shown in figure 2 for $\lambda = 0, \lambda^*$. The errors are indeed tiny and the modular crossing (58) is numerically satisfied to good accuracy.

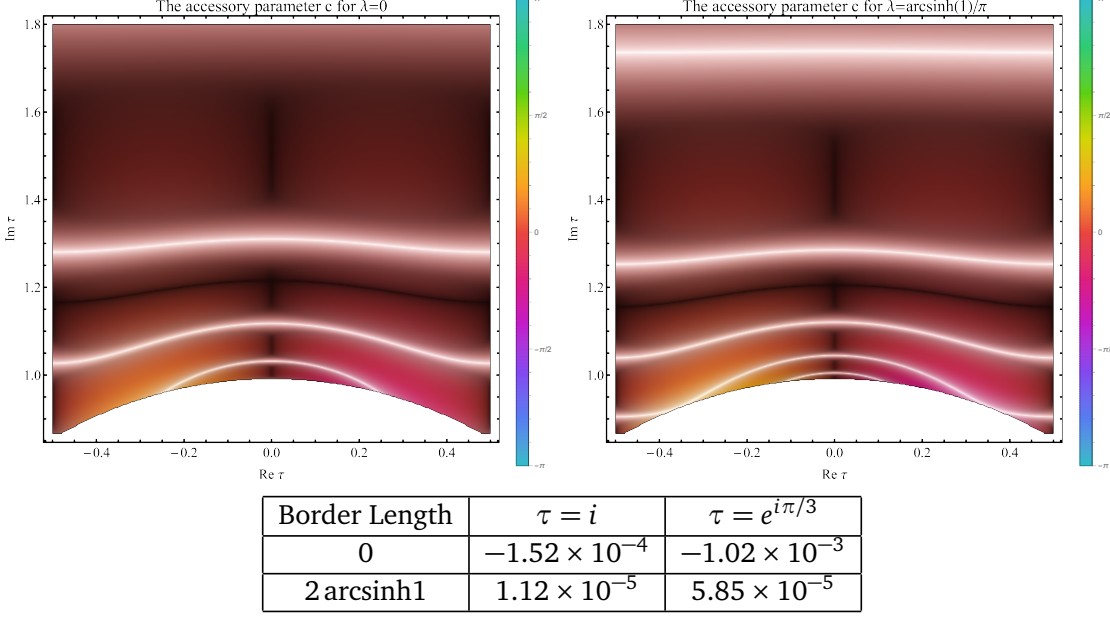

| Border Length | $\tau = i$ | $\tau = e^{i\pi/3}$ |
|---|---|---|
| 0 | $-1.52 \times 10^{-4}$ | $-1.02 \times 10^{-3}$ |
| $2\,\mathrm{arcsinh}1$ | $1.12 \times 10^{-5}$ | $5.85 \times 10^{-5}$ |

Figure 3: The accessory parameter $c = c(\tau, \overline{\tau})$. The black contours are for the real and imaginary parts, white contours are for the absolute value, and the color shading indicates the phase in the figure. We show the result we obtained for $\tau = 0, e^{i\pi/3}$ in the table blow. Recall $c = 0$ for them.

Let us finally consider the expansion of the accessory parameter $c = c(q, \overline{q})$ (41) in moduli. Observe that we have, from (49)

$$4\pi^2 q \frac{\partial}{\partial q} = 4\pi^2 q \frac{\partial \xi}{\partial q} \frac{\partial}{\partial \xi} = 2\pi \xi^2 \frac{\partial}{\partial \xi}. \tag{59}$$

Endowed with this, we can find the perturbative expansion for the accessory parameter using (25c) and (57). This is given by

$$c(q, \overline{q}; \lambda) = \pi^2 \left[ \frac{(1 + \lambda^2)}{6} - (1 + \lambda^2)(5 + \lambda^2) q + \cdots \right] + \pi^2 \left[ -2 + (1 + \lambda^2)^2 q + \cdots \right] \xi^2 + \cdots \tag{60}$$

This series is invariant under involution symmetry (15) by construction. We note that the coefficient of $\xi^0$ is a holomorphic function of $q$, despite the entire series is not. This point will be important in the next subsection when we investigate the asymptotics of this formula.

The overall behavior of the accessory parameter $c = c(\tau, \overline{\tau})$ in $\mathcal{M}_{1,1}(L)$ is shown in figure 3. It is apparent from the figures that the accessory parameter is indeed involution symmetric and we obtained results closer to the exact values whenever available. The investigation of the modularity of the accessory parameter (60) is relegated to appendix C.

Given (60), it is possible to investigate the degeneration limit $q \to 0$ ($\tau \to i\infty$) analytically, albeit it is not relevant for CSFT. We see $c \to (1 + \lambda^2)\pi^2/6 = \delta \pi^2/3$ from (60). On top of this, the Weierstrass elliptic function takes the form

$$\lim_{\tau \to i\infty} \wp(z, \tau) = \sum_{n \in \mathbb{Z}} \left[ \frac{1}{(z - n)^2} - \frac{1}{n^2} \right] = \pi^2 \csc^2(\pi z) - \frac{\pi^2}{3}, \tag{61}$$

as the sum over the lattice $\Lambda$ reduces to just the sum of the real lattice points. In the second line we evaluated the infinite sum using an identity and the Riemann zeta function. As a result,

$T(z)$ in the Lamé equation (12) evaluates to

$$T(z) = \delta \, \pi^2 \csc^2(\pi z). \tag{62}$$

In the limit $\tau \to i\infty$, we have a punctured infinite strip in the $z$-plane due to identification $z \sim z + 1$ and the punctures are placed at $z = 0$ and its images. This geometry can be conformally mapped to the thrice-punctured sphere $u$ with the exponential map $u = \exp(2\pi i z)$. The puncture at $z = 0$ (and its images) gets mapped to $u = 1$ and $z = \pm i\infty$ are mapped to $u = 0, \infty$ respectively. Because of the degeneration limit, the classical weights associated with $u = 0, \infty$ are supposed to taken to be $1/2$ (i.e they are genuine cusps). These points are identified with each other to create a noded once-bordered torus.

Now recall that $\widetilde{T}(u)$ in the Fuchsian equation for the three-punctured sphere is given by [37]

$$\widetilde{T}(u) = \frac{\delta_1}{u^2} + \frac{\delta_2}{(1-u)^2} + \frac{\delta_1 + \delta_2 - \delta_3}{u(1-u)}. \tag{63}$$

Pulling back $\widetilde{T}(u)$ to the $z$-plane with the exponential map $u = \exp(2\pi i z)$ using (3), evaluating $\{u, z\} = 2\pi^2$, and subsequently taking $\delta_1 = \delta, \delta_2 = \delta_3 = 1/2$ it can be shown that it indeed produces (62). We see that our procedure generates a sensible result in the degeneration limit.

## 3.3 The Weil-Petersson metric and the volume of $\mathcal{M}_{1,1}(L)$

In this subsection, we solve for the Weil-Petersson (WP) metric $g_{WP}$ on the moduli space $\mathcal{M}_{1,1}(L)$ as a series expansion in moduli and compute its associated volumes. A mathematically oriented introduction to the WP metric can be found in [41].

We *claim* that the WP metric $g_{WP} = g_{\tau\bar{\tau}}|d\tau|^2 = g_{q\bar{q}}|dq|^2$ on $\mathcal{M}_{1,1}(L)$ is given by

$$g_{\tau\bar{\tau}} = A \partial_{\bar{\tau}} c = -2\pi i A \partial_{\bar{\tau}} \partial_{\tau} S_{HJ}^{(1,1)}(\tau, \bar{\tau}; \lambda)$$

$$\implies g_{q\bar{q}} = -2\pi i A \, \partial_{\bar{q}} \partial_q S_{HJ}^{(1,1)}(q, \bar{q}; \lambda) = \frac{A}{2\pi i \, |q|^2} \, \bar{q} \, \partial_{\bar{q}} c(q, \bar{q}; \lambda), \tag{64}$$

where $A$ is a complex constant (with a possible dependence on $\lambda$) that we are going to determine momentarily. In other words, we claim that the on-shell action $S_{HJ}^{(1,1)}$ is essentially the Kähler potential for $g_{WP}$. As far as the knowledge of the author goes such a relation hasn't been proven and we are not going to argue for it, as it would take us far from the scope of this work. Instead we will just explore its consequences. Still, it may be possible to argue this relation using the conformal Ward identity heuristically just as in (23), but with two stress-energy insertions, along the lines discussed in [60–62] and this was our motivation behind our claim in (64). We remark that in the case of genus 0 with elliptic/parabolic singularities an analogous relation has already established rigorously [63] and tested for the four-punctured sphere in [45]. Clearly the metric (64) is invariant under modular transformations.

Let us begin by fixing the constant $A$. We do this by investigating the degeneration limit $\tau \to i\infty$ ($q \to 0$). In this limit it is easy to see that the length of the internal geodesic $\ell$ and twist $\theta$ it goes under (with $\ell \sim \ell + \theta$ describing the same surface) asymptotically takes the form

$$\ell = 2\pi\lambda_s \approx -\frac{4\pi^2}{\log|q|^2}, \qquad \frac{2\pi\theta}{\ell} \approx \arg q, \tag{65}$$

from (49) and (53). Using the Wolpert's magic formula [64], the WP 2-form $\omega_{WP}$ associated with the WP metric asymptotically becomes

$$\omega_{WP} = d\ell \wedge d\theta \approx -\frac{16\pi^3}{|q| \log^3 |q|^2} \, d(|q|) \wedge d(\arg q) = -\frac{8\pi^3 i}{|q|^2 \log^3 |q|^2} dq \wedge d\bar{q}, \tag{66}$$

and from this we see that the WP metric is

$$\omega_{WP} = \frac{i}{2} g_{q\bar{q}} \, dq \wedge d\bar{q} \implies g_{q\bar{q}} \approx -\frac{16\pi^3}{|q|^2 \log^3 |q|^2}, \tag{67}$$

asymptotically. Given the equations (59) and (60), we also find the WP metric asymptotically to be, through (64),

$$g_{q\bar{q}} \approx \left( \frac{A}{2\pi i \, |q|^2} \right) \times \left( \frac{16\pi^4}{\log^3 |q|^2} \right) \approx -\frac{8\pi^3 i A}{|q|^2 \log^3 |q|^2} \implies A = -2i. \tag{68}$$

The first term was due to the normalization in (64) while the second term is from (60). Note that the coefficient of the $\xi^0$ term in (60) being holomorphic was crucial to have this asymptotic form. In summary, we conclude that $A = -2i$ from last two relations and we find it is independent of the length of the border. We emphasize that it should be possible to obtain this normalization factor by deriving the identity (64) from the conformal Ward identity. Regardless, it is already promising that two distinct methods of calculating the WP metric close to degeneration yields the same result. In fact, this is exactly the expected asymptotic form [65–67].

Equipped with the normalization, we can use (64) to write the WP metric as a series expansion

$$g_{q\bar{q}} = \frac{\xi^3}{|q|^2} \left[ \left( 2 - 2(1 + \lambda^2)^2 \mathrm{Re}(q) + \cdots \right) + \left( -\frac{6(1 + \lambda^2)^2}{\pi} \mathrm{Re}(q) + \cdots \right) \xi + \cdots \right]. \tag{69}$$

An important point to notice here that this metric has to be real by (64) and its normalization. We see this is indeed the case: the non-perturbative corrections always combine with each other to have a dependence on the real part of $q$ exclusively.

Given $g_{WP}$, it is possible to compute the WP volume $V_{WP}$ of $\mathcal{M}_{1,1}(L)$ by

$$V_{WP}(L) = \int_{\mathcal{M}_{1,1}(L)} \omega_{WP} = \frac{i}{2} \int_{\mathcal{M}_{1,1}(L)} g_{\tau\bar{\tau}} \, d\tau \wedge d\bar{\tau}. \tag{70}$$

This volume has an exact expression and it is given by [68]

$$V_{WP}(L) = \frac{\pi^2}{6} + \frac{L^2}{24} \approx 1.645 + 0.0417 L^2, \tag{71}$$

in our conventions. We compute the volume $V_{WP}$ using (69) and the results are shown in figure 4. We performed Monte-Carlo (MC) integration by uniformly sampled $10^5$ points in the fundamental region with a cutoff placed at $\mathrm{Im}(\tau) = 20$ to evaluate (70). For each value of $L$, we have repeated the integration 10 times and take the mean. As shown in figure 4, we have a satisfactory match and the quadratic dependence of the volume to the length of the border is apparent. This result strongly suggests the formula (64) produces the WP metric on $\mathcal{M}_{1,1}(L)$.

## 4 The vertex and Feynman regions

In this section we consider the case with $s > 0$ in (38), which is relevant for the geometries that contain internal flat cylinders, i.e. string propagators. This will subsequently lead to the description of the boundary of the vertex region $\partial \mathcal{V}_{1,1}(L)$ and the dependence of the Schwinger

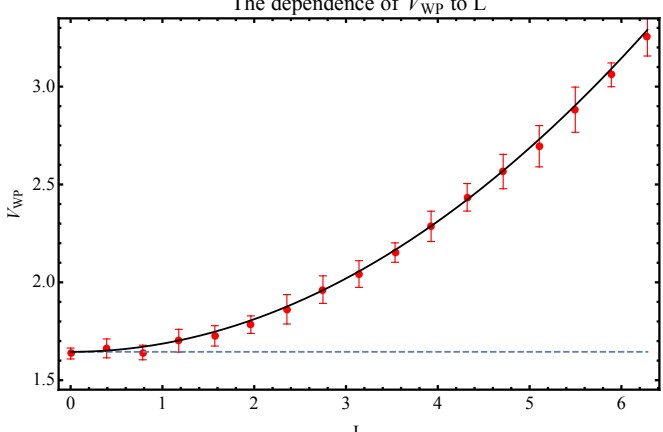

Figure 4: The volume $V_{WP}(L)$ as a function of the length of the border $L$. The solid curve is the exact result (71), the dashed curve is $\pi^2/6$ and the red points are the results of our integration. The uncertainties are due to MC integration. The best fit weighted by the uncertainties is given by $V_{WP}(L) = 1.635 + 0.0413L^2$, which is sufficiently close to the analytic result (71).

parameter of the propagator $q = e^{-s+i\theta}$ to the moduli $\tau$ and the length of the border $L$ of the torus.

Begin by considering the saddle-point equation (39) again, but with $s > 0$

$$s = -\log|q| = \frac{1}{2\lambda_s} \frac{\partial}{\partial \lambda'} \left[ S_{HJ}^{(0,3)}(\lambda', \lambda, -\lambda') - 2f_{\lambda'}^{\lambda}(q) - 2\overline{f}_{\lambda'}^{\lambda}(\overline{q}) \right]_{\lambda'=\lambda_s} = 0. \tag{72}$$

Given the moduli $\tau$ and the border length $L$, the length of the internal geodesic $2\pi\lambda_s$ and $s$ are not independent from each other. Since the circumference of the string propagator in hyperbolic CSFT is given by $2\pi\lambda$, we consider the situation where we set $\lambda_s = \lambda$ in this subsection.

We would like to evaluate $s$ as a function of the moduli $\tau$ and the length of the border $L$. For that we need the following two derivatives. The first one is the derivative of the action $S_{HJ}^{(0,3)}$ and the classical torus conformal blocks evaluated at $\lambda' = \lambda$. We find

$$s = -\log|q| = 2\log R(\lambda, \lambda, -\lambda) - \frac{2}{\lambda} \mathrm{Re} \left[ \frac{\partial f_{\lambda'}^{\lambda}(q)}{\partial \lambda'} \right]_{\lambda'=\lambda} \tag{73}$$

$$= -\frac{\pi}{\lambda} + \frac{i}{\lambda} \log \left[ \gamma\left(\frac{1}{2} + \frac{3i\lambda}{2}\right) \gamma\left(\frac{1}{2} + \frac{i\lambda}{2}\right) \frac{\Gamma(1-i\lambda)^2}{\Gamma(1+i\lambda)^2} \right] - \frac{1}{2}\log|q|^2 + \frac{1}{2}\mathrm{Re}\, q + \cdots$$

The boundary of the vertex region is placed at $s = 0$ and this produces the following curve for $\partial\mathcal{V}_{1,1}(L)$ (notice the similarity to the form given in [40])

$$R(\lambda, \lambda, -\lambda)^{\lambda} = e^{-\pi/2} \left[ \gamma\left(\frac{1}{2} + \frac{3i\lambda}{2}\right) \gamma\left(\frac{1}{2} + \frac{i\lambda}{2}\right) \frac{\Gamma(1-i\lambda)^2}{\Gamma(1+i\lambda)^2} \right]^{1/2i} = \left| \exp\left[ \frac{\partial f_{\lambda'}^{\lambda}(q)}{\partial \lambda'} \right] \right|_{\lambda'=\lambda}. \tag{74}$$

The shape of this curve for assorted values of $\lambda$ has been plotted in figure 5. Note that the boundary curves $\partial\mathcal{V}_{1,1}(L)$ may appear like constant $\mathrm{Im}(\tau)$ lines, but this is just an illusion of $\mathrm{Re}(q^n)$ terms in classical torus blocks being highly suppressed when $\mathrm{Im}(\tau) \gtrsim 1/2$. We observed decreasing the value of $\lambda$ decreases the size of the Feynman region. This makes sense, given

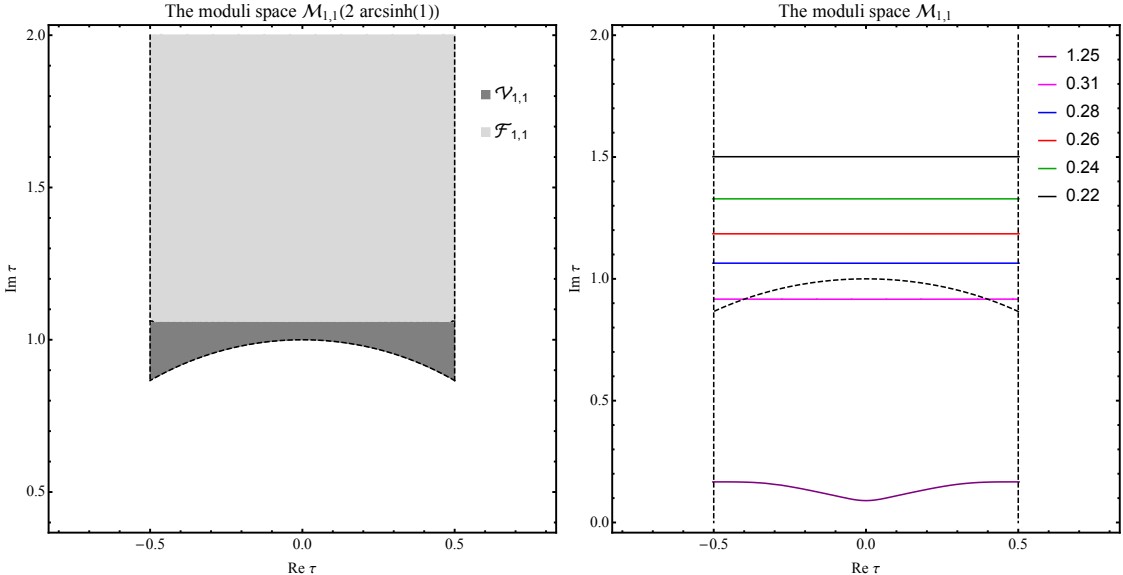

Figure 5: Left: The decomposition of the moduli space $\mathcal{M}_{1,1}(L^*)$ to the vertex $\mathcal{V}_{1,1}(L^*)$ and the Feynman $\mathcal{F}_{1,1}(L^*)$ regions. Right: The progression of the boundary curve $\partial\mathcal{V}_{1,1}(2\pi\lambda)$ as a function of $\lambda$. As $\lambda$ increases, the vertex region shrinks and eventually disappears.

that $\lambda \to 0$ the entire moduli space turns into the vertex region since the length of the internal geodesics is always larger than the boundary length [34].

There appears to be two "critical" values for $\lambda$: the values when the boundary touches $\tau = i$ and $\tau = e^{i\pi/3}$. Let us name them $\lambda_1$ and $\lambda_2$ respectively. We estimated $\lambda_1 \approx 0.292$ and $\lambda_2 \approx 0.332$. When $0 < \lambda < \lambda_1$, as in the case of quantum hyperbolic CSFT, the vertex and Feynman regions are present and the Feynman region is covered once (see figure 6). When $\lambda_1 < \lambda < \lambda_2$, the vertex and Feynman region are still present, however the part of the Feynman region gets covered finitely many times, as we have to map the Schwinger parameter q to the outside of the fundamental region. Once this part gets mapped back to the fundamental region it leads to overcounting. Finally, we have $\lambda > \lambda_2$, for which the vertex region disappears entirely whereas the Feynman region is still covered multiple times.

Looking at the $\lambda \to \infty$ limit is also interesting. In this case the hyperbolic three-string vertex with two punctures got sewed reduces to the minimal-area vertex described by a Strebel differential with the same punctures got sewed together [40]. This is precisely the situation investigated three decades ago in [69] and it is argued that the Feynman region gets covered infinitely-many times.[4] Our claim is that the curve (74), in the $\lambda \to \infty$ limit, produces the behavior for the same curve shown in figure 10 of [69]. We have qualitatively observed that $\partial\mathcal{V}_{1,1}(L)$ in (74) appears to approach the behavior given in [69] as $L$ increases, see figure 5.

We point out that it is difficult to investigate the behavior around $|q| \approx 1$ (i.e. $\mathrm{Im}(\tau) \approx 0$) directly by taking the $\lambda \to \infty$ limit of (74), even though we have observed such WKB-like limit appears to exist for the classical torus blocks of [42], akin to the case of classical 4-point blocks [40]. This can be attributed to the facts that the torus conformal block (B.1) converges uniformly on the regions $\{q, |q| = e^{-\epsilon} < 1\}$ for any $\epsilon > 0$ (which is apparent following the reasoning in [70]) and the series seem to require increasing number of terms in q for smaller $\epsilon$.

---

[4]The author thanks Barton Zwiebach for pointing out this reference.

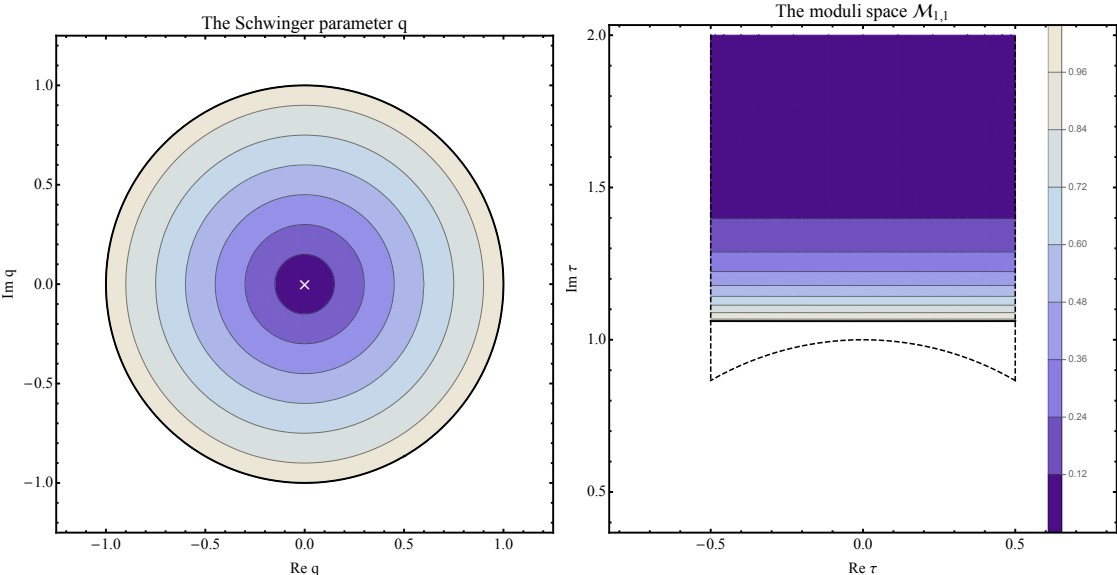

Figure 6: The behavior of the Schwinger parameter $\mathfrak{q} = \mathfrak{q}(\tau)$ (75) in the Feynman region $\mathcal{F}_{1,1}(L^*)$.

Finally, we can find the Schwinger parameter of the string propagator $\mathfrak{q}$ as a function of the moduli $\tau$ and the length of the border $L$ using (73). This is given by

$$
\mathfrak{q}(\tau;\lambda) = e^{\pi/\lambda} \left[ \gamma\left(\frac{1}{2} + \frac{3i\lambda}{2}\right) \gamma\left(\frac{1}{2} + \frac{i\lambda}{2}\right) \frac{\Gamma(1 - i\lambda)^2}{\Gamma(1 + i\lambda)^2} \right]^{-1/i\lambda} \exp\left[ \frac{2}{\lambda} \frac{\partial f_{\lambda'}^{\lambda}(q)}{\partial \lambda'} \right]_{\lambda'=\lambda}
$$

$$
= R(\lambda, \lambda, -\lambda)^{-2} \exp\left[ \frac{2}{\lambda} \frac{\partial f_{\lambda'}^{\lambda}(q)}{\partial \lambda'} \right]_{\lambda'=\lambda}. \tag{75}
$$

The argument for this is just as in [40]. The parameter $\mathfrak{q}$ is a holomorphic function of the moduli $\tau$ (therefore $q$) for $0 < |\mathfrak{q}| \leq 1$ for which the curve $|\mathfrak{q}| = 1$ describes $\partial \mathcal{V}_{1,1}(L)$. Furthermore, the point $\mathfrak{q} = 0$ has to get mapped to the boundary of the moduli space $\tau = i\infty$ (i.e $q = 0$) with a well-defined Taylor expansion [69, 71]. This fixes the function above uniquely (up to an unimportant phase) by the Riemann mapping theorem. The behavior of this function for $\lambda = \lambda^*$ is shown in figure 6.

## 4.1 The accessory parameter for Feynman diagrams

In this subsection we modify the arguments of the previous section to derive the accessory parameter for the situation where there is a string propagator present in the geometry. Calling the on-shell action resulting from (40) $S_{HJ,s>0}^{(1,1)}(q,\overline{q};\lambda)$ when $s > 0$ and defining

$$
S_{HJ}^{(1,1),\mathcal{F}}(q,\overline{q};\lambda) \equiv S_{HJ}^{(1,1),s>0}(q,\overline{q};\lambda) + \lambda^2 s, \tag{76}
$$

the relevant accessory parameters is given by, upon replacing $S_{HJ}^{(1,1)} \to S_{HJ}^{(1,1),\mathcal{F}}$ in (41),

$$
c(q;\lambda) = (1 + \lambda^2)\, \eta_1(q) + 4\pi^2 q\, \frac{\partial S_{HJ}^{(1,1),\mathcal{F}}}{\partial q}(q,\overline{q};\lambda). \tag{77}
$$

The justification for this can be provided as in [40]: there is an annulus part in the geometry so there should be additional contributions to the on-shell action due to the modulus of the cylinder, while the rest of the on-shell action does not get affected.

Using (40), we immediately see

$$S_{HJ}^{(1,1),\mathcal{F}}(\tau,\overline{\tau};\lambda) = S_{HJ}^{(0,3)}(\lambda,\lambda,-\lambda) - 2f_\lambda^\lambda(q) - 2\overline{f}_\lambda^\lambda(\overline{q}) \tag{78}$$

$$= 2F\left(\frac{1}{2}+\frac{3i\lambda}{2}\right) + 6F\left(\frac{1}{2}+\frac{i\lambda}{2}\right) + 6H(i\lambda) + 3\pi\lambda - 2f_\lambda^\lambda(q) - 2\overline{f}_\lambda^\lambda(\overline{q}),$$

where we have used (35), together with identities $F(1/2+x) = F(1/2-x)$ and $H(x) = H(-x)$, see (36). From this, the accessory parameter for the Feynman region $c_{\mathcal{F}}$ is given by

$$c_{\mathcal{F}}(q;\lambda) = (1+\lambda^2)\,\eta_1(q) - 8\pi^2 q\,\frac{\partial f_\lambda^\lambda(q)}{\partial q} = \delta\frac{\pi^2}{3} - 2\pi^2\lambda^2 - 5\pi^2(1+\lambda^2)q + \mathcal{O}(q^2), \tag{79}$$

which is a holomorphic function in the moduli as somewhat expected.

Similar to what we did in subsection (3.2), an interesting limit to investigate is the degeneration limit, i.e. $\tau \to i\infty$ or $q \to 0$. In this limit we have $c_{\mathcal{F}} \to \delta\pi^2/3 - 2\pi^2\lambda^2$. Given this and the identity (61), $T(z)$ in the Lamé equation evaluates to

$$T(z) = \delta\,\pi^2\csc^2(\pi z) - 2\pi^2\lambda. \tag{80}$$

Observe how $\delta\pi^2/3$ part of $c_{\mathcal{F}}$ coming from $\eta_1$ has canceled. When $\lambda \to 0$, this is equal to (62). This is consistent, the flat cylinder degeneration reduces to the cusp-like degeneration when $\lambda = 0$.

Like earlier, the punctured infinite strip in the $z$-plane can be conformally mapped to the three-punctured sphere $u$ with the exponential map $u = \exp(2\pi i z)$. Because of the flat cylinder degeneration, we identify the *holes* around $u = 0$ and $u = \infty$ this time. Accordingly, we take $\delta_1 = \delta_2 = \delta_3 = \delta$ in (63). After pulling $\widetilde{T}(u)$ to the $z$-plane, we indeed obtain (80). This result supports the validity of (79). For the general cases, we should compare (79) with the accessory parameter obtained from sewing the pair-of-pants with itself. Unfortunately, the description of the latter is not currently available.

Given the accessory parameter (79), it is possible to find the local coordinates for the surfaces in the Feynman region $\mathcal{F}_{1,1}(L)$ using the Lamé equation as well. This is the task we describe in the next section, along with finding the local coordinates for the vertex region $\mathcal{V}_{1,1}(L)$.

# 5 The local coordinates

Finally, we describe the procedure to derive the local coordinate for the hyperbolic tadpole vertex and the one-loop Feynman diagrams. We have already found the accessory parameters solving the hyperbolic monodromy problem through the Polyakov conjecture for both, so all we have to do is to solve the Lamé equation (12) and relate its solutions to the local coordinates. In the first subsection we describe the Lamé function that will be used to construct the local coordinates. Then we make some remarks on the computation of the mapping radii in the subsequent subsection and derive the local coordinates in the final subsection.

## 5.1 Lamé functions

The solutions to the Lamé equation are called *Lamé functions* and we are going to consider their expansions around $z = 0$. Suppose they have the expansion of the form

$$\psi(z) = z^\alpha \sum_{n=0}^\infty a_n z^n, \quad \text{with} \quad a_0 = 1, \tag{81}$$

where $\alpha$ is a complex number. Considering the series (A.3), we arrive to the following equality

$$\sum_{n=0}^{\infty} a_n(n+\alpha)(n+\alpha-1)z^{n-2} + \left[\frac{c}{2} + \frac{\delta}{2z^2} + \frac{\delta}{2}\sum_{n=1}^{\infty}(2n+1)G_{2n+2}z^{2n}\right]\left[\sum_{n=0}^{\infty}a_n z^n\right] = 0. \quad (82)$$

Let us focus on the leading term $\sim z^{-2}$. This gives

$$\alpha = \frac{1}{2}(1 \pm i\lambda), \quad (83)$$

essentially by our construction. This choice realizes a diagonal real hyperbolic monodromy around the puncture $z = 0$. Different sign choices in $\alpha$ leads to two linearly independent solutions.

The rest of the expansion can be found by matching powers of $z$ with each other and solving for the coefficients $a_n$ recursively. For example, it is easy to see that $a_1 = 0$ for the next term from the coefficient of $z^{-1}$ (in fact all odd powers of $z$ vanishes by symmetry). We find

$$\psi^{\pm}(z) = \frac{z^{(1\pm i\lambda)/2}}{\sqrt{i\lambda}}\left[1 - \frac{c}{4(-2\pm i\lambda)}z^2 - \frac{c^2 - 6(2\pm i\lambda)(1+\lambda^2)G_4}{32(-8 + \lambda(\lambda \mp 6i))}z^4 + \cdots\right]. \quad (84)$$

Here we have included an overall multiplicative factor to normalize the Wronskian to one, $W(\psi^-, \psi^+) = 1$. We point out that the overall phase of $z$ itself is ambigious while solving (82). This is not to be confused with the irrelevant overall phase of the local coordinates of CSFT. We are going to comment on how this phase ambiguity can be resolved momentarily.

The associated scaled ratio is given by

$$\rho(z) = \left(\frac{\psi^+(z)}{\psi^-(z)}\right)^{1/i\lambda} = z\left[1 + \frac{c}{2(4+\lambda^2)}z^2 + \frac{36c^2 + 3(1+\lambda^2)(4+\lambda^2)^2 G_4}{8(4+\lambda^2)^2(16+\lambda^2)}z^4 + \cdots\right]. \quad (85)$$

We emphasize the dependence on the moduli $\tau$ appears in the accessory parameter $c$ and the Eisenstein series $G_{2n}$, see (A.4). This series is expected to converge until $z$ hits an image of the puncture at $z = 0$, but it is possible to analytically continue beyond it. Notice the hyperbolic monodromy for these solutions is not compatible yet–it will be upon including the correct mapping radii, which we undertake next subsection.

## 5.2 Mapping radii

Like in [37], the local coordinates are related to the scaled ratio up to a multiplicative constant that depends on $\lambda$, whose inverse is the mapping radius associated with the local coordinate. Recall that the definition and transformation of the mapping radius is given by

$$r(z) = \left|\frac{dz}{dw}\right|_{w=0}, \quad \text{and} \quad r(z) = \left|\frac{\partial\widetilde{z}}{\partial z}\right|^{-1} r(\widetilde{z}). \quad (86)$$

This quantity has determined by the pair of hypergeometric functions ${}_2F_1(a, b; c; z)$ realizing the hyperbolic monodromy around each puncture and demanding compatible monodromies in the case of hyperbolic three-string vertex. Such connection formulas are not available for the Lamé functions generally. Despite this obstacle, it is possible to obtain the mapping radii like it is done for the four-bordered spheres in [40].

Before we discuss the derivation of the mapping radius for once-bordered tori, we expand the discussion in [40, 50] and argue for the following identity for a Riemann surface $\Sigma_{g,n}$ and the on-shell action $S_{HJ}^{(g,n)}[\varphi]$ associated with such surface:

$$\frac{\partial S_{HJ}^{(g,n)}[\varphi]}{\partial\lambda_i} = 2\lambda_i \log r_i(u), \quad (87)$$

where $H_i$ is the flat region around $i$th puncture ("hole") and $r_i(u)$ is its associated mapping radius in the $u$-plane. This is the plane for which the surface $\Sigma_{g,n}$ is uniformized as the Riemann sphere containing $n+2g$ punctures and the holes around $g$ punctures are identified with the plumbing fixture in their local coordinates to account for the genus. The geodesic length of $\partial H_i$ is given by $2\pi\lambda_i$ as usual. The distinguishing feature of this plane is that it is like the complex plane considered in [40], but addition of appropriate identifications. Given this, the calculus in such plane is simple and we will be able to do the operations described below. Finally, Note that we are taking *partial* derivative with respect to one of the $\lambda_i$'s while keeping the others fixed in (87).

In the $u$-plane, upon taking derivative with respect to $\lambda_i$, integrating-by-parts and employing the equation of motion, the only remaining terms are the ones associated with the holes $H_i$ around the punctures. This is because after these operation we are left with terms localized on the boundary of the patches after these operation and sewing them cancel each other, localizing the terms to the small regularization circles around each puncture. This reasoning here also restricts the appearance of the derivative of the mapping radii with respect to $\lambda_i$, see the form given in equation (B.2) of [40] and equation (23) in [50]. In the view of this remark, let us place the $i$th puncture at $u = 1$ and write

$$\frac{\partial S_{HJ}^{(g,n)}[\varphi]}{\partial \lambda_i} = \lim_{\epsilon \to 0} \left[ \frac{i}{4\pi} \sum_{j=1}^{n} \int_{|u-1|=\epsilon} \frac{\partial \varphi}{\partial \lambda_i} \left( \frac{\partial \varphi}{\partial \overline{u}} d\overline{u} - \frac{\partial \varphi}{\partial u} du \right) + \frac{1}{2\pi\epsilon} \sum_{j=1}^{n} \int_{|u-1|=\epsilon} |du| \frac{\partial \varphi}{\partial \lambda_i} \right]$$
$$+ 2\lambda_i \log r_i(u). \tag{88}$$

Furthermore, we have the Weyl factor around the $i$th puncture as the following expansion on $\mathbb{C}H_i$

$$\varphi(u,\overline{u}) = \log \lambda_i^2 - \log |u-1|^2 + \mathcal{O}(u-1, \overline{u}-1), \tag{89}$$

since it describes flat semi-infinite cylinder around $u = 1$. As a result, the only remaining term in (88) are those associated with the $i$th puncture. But a quick observation shows that these two terms in (88) cancel each other and we indeed obtain (87).

We emphasize again that the mapping radius $r_i(u)$ above is given in the $u$-plane. Depending on the way surface $\Sigma_{g,n}$ is uniformized, we have to transform the mapping radii according to (86). In the case of $g = n = 1$, we would like the mapping radius $r(z) = r$ for the $z$-plane and using the exponential map $u = e^{2\pi i z}$ we actually have to use the relation

$$\frac{\partial S_{HJ}^{(1,1)}}{\partial \lambda}(\tau, \overline{\tau}; \lambda) = 2\lambda \log 2\pi r, \tag{90}$$

given $r_i(u) = 2\pi r(z) = 2\pi r$ by (86).

As a consistency check, let us demonstrate the change in both sides of (90) is the same under the modular transformations (10). The $T$ portion of the modular transformation is trivial. For the $S$ portion, the change of the right-hand side of (87) is given by, from the definition of the mapping radius (86),

$$r\left(\frac{z}{\tau}, -\frac{1}{\tau}\right) = \frac{r(z,\tau)}{|\tau|} \implies 2\lambda \log r\left(\frac{z}{\tau}, -\frac{1}{\tau}\right) = 2\lambda \log r(z,\tau) - 2\lambda \log |\tau|. \tag{91}$$

This is exactly how the left-hand side of (87) changes by (32) upon taking derivative with respect to $\lambda$. So we see they are indeed consistent.

We would like to take a derivative of $S_{HJ}^{(1,1)}(\tau, \overline{\tau}; \lambda)$ with respect to $\lambda$ now. We first notice is the parameter $\xi$ of (49) depends on $\lambda$ and we have

$$\frac{\partial \xi}{\partial \lambda} = -\frac{\xi^2}{2\pi} \frac{ds_1(\lambda)}{d\lambda} = -\frac{2\xi^2}{\pi} \operatorname{Im} \psi^{(1)}\left(\frac{1}{2} + \frac{i\lambda}{2}\right). \tag{92}$$

In the view of this, we find

$$\log 2\pi r = \left[ -\frac{\pi}{2\lambda} + \frac{i}{2\lambda} \log\left[ \gamma\left(\frac{1}{2} + \frac{i\lambda}{2}\right)^4 \left(\frac{\Gamma(1-i\lambda)}{\Gamma(1+i\lambda)}\right)^2 \right] - (1+\lambda^2)\text{Re}(q) + \cdots \right] + \cdots \quad (93)$$

We have adjusted branches in the expression above in order to match the form given in [37]. Notice that when $q \to 0$ this generates the mapping radius derived in [37], more specifically $R(0, \lambda, 0)$ in (43). Also notice as $\lambda \to 0$, we have

$$2\pi r \approx 16 \exp\left(-\frac{\pi}{2\lambda}\right)[1 + \cdots], \quad (94)$$

which is the expected behavior from the appearance of a hyperbolic cusp [34, 37]. Here dots indicate the subleading terms in $\lambda$.

The identity analogous to (87) applies to the situation when the internal flat cylinders are present. Imagine for each $3g - 3 + n$ disjoint simple closed geodesics of length $2\pi\lambda_k$ we have grafted flat cylinder of size $s_k$. This will modify the on-shell action by

$$S_{HJ}^{(g,n),\mathcal{F}}[\varphi] \equiv S_{HJ}^{(g,n),s>0}[\varphi] + \sum_{k=1}^{3g-3+n} \lambda_k^2 s_k. \quad (95)$$

Here $\lambda_k$'s of the internal geodesics are distinct from those of the borders. The identity (87) still holds in this situation upon replacing $S_{HJ}^{(g,n)}[\varphi] \to S_{HJ}^{(g,n),\mathcal{F}}[\varphi]$ by the similar reasoning before. In the case of $g = n = 1$, this can be further written as

$$\frac{dS_{HJ}^{(1,1),\mathcal{F}}}{d\lambda}(\tau, \overline{\tau}; \lambda) = 2\lambda \log 2\pi r_\mathcal{F} + 2\lambda s. \quad (96)$$

Here $r_\mathcal{F}$ stands for the mapping radius in the Feynman region in the $z$-plane. Note that we take a *total* derivative with respect to $\lambda$ in this expression. This is the form we are going to consider due to simplicity of (78). The mapping radii $r_\mathcal{F}$ is then given by

$$\log 2\pi r_\mathcal{F} = \frac{i}{2\lambda} \log\left[ -\frac{\pi}{2\lambda} \gamma\left(\frac{1}{2} + \frac{3i\lambda}{2}\right) \gamma\left(\frac{1}{2} + \frac{i\lambda}{2}\right) \left(\frac{\Gamma(1-i\lambda)}{\Gamma(1+i\lambda)}\right)^2 \right] - \text{Re}(q) + \cdots, \quad (97)$$

using (78). Again, the branches are adjusted appropriately for this expression. In the $q \to 0$ limit we generate the mapping radius $R(\lambda, \lambda, -\lambda)$ in (43) of [37], as it is expected from this limit. The modularity concerns work similarly to (91) and the $\lambda \to 0$ behavior matches with (94).

### 5.3 The local coordinates and the hyperbolic metric

Endowed with the mapping radii we can obtain the local coordinate patch on the one-punctured torus for given $\tau \in \mathcal{M}_{1,1}$ and $\lambda$ using (60) and (79). We have

$$\widetilde{w}(z) = \frac{\rho(z)}{r} = \frac{z}{r}\left[ 1 + \frac{c}{2(4+\lambda^2)}z^2 + \frac{36c^2 + 3(1+\lambda^2)(4+\lambda^2)^2 G_4}{8(4+\lambda^2)^2(16+\lambda^2)}z^4 + \cdots \right], \quad (98)$$

where $r, c$ are the associated mapping radii in the $z$-plane and the accessory parameter corresponding to the vertex or Feynman regions, respectively.

The function $\tilde{w}(z)$ is not exactly the local coordinate for the hyperbolic tadpole as there was a phase ambiguity in $z$ mentioned below (84), which has been propagated here. The ambiguity in question can be set by letting

$$w(z) = \widetilde{w}\left(ie^{-i\arg\tau}z\right), \quad (99)$$

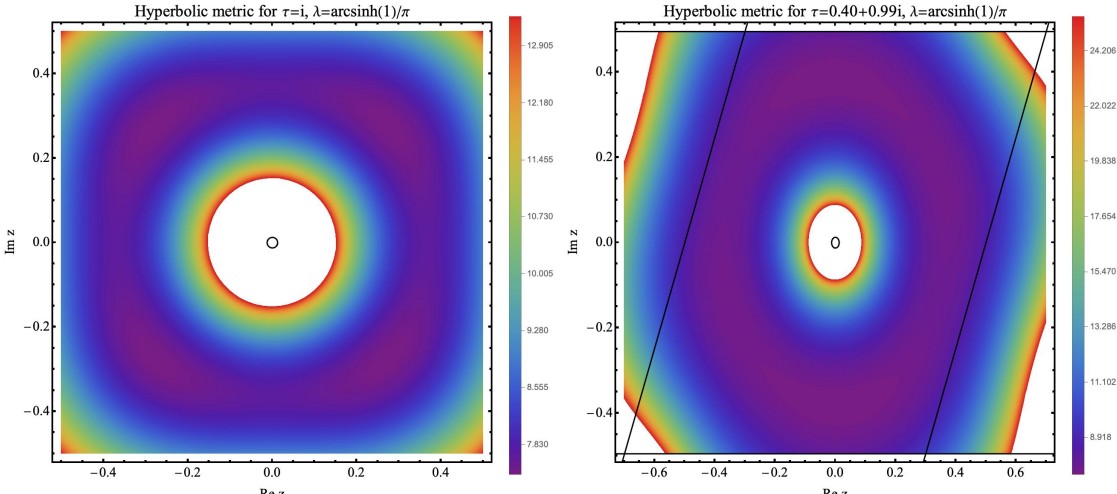

Figure 7: The local coordinate patch (inside the black curve) and the hyperbolic metric for the bordered tori $\tau = i$ and $\tau = 0.40 + 0.99i$. The length of the geodesics border is $L = L^* = 2 \operatorname{arcsinh} 1$. Constant $e^\varphi$ contours have plotted. We have used $\mathcal{O}(z^{93})$ in the expansion (99).

which would produce the true local coordinates. This is the correct choice, given that when $\tilde{w}(z)$ is complex conjugated it describes the local coordinate for the torus whose moduli is $\bar{\tau}$. On the other hand, when $w(z)$ is complex conjugated it still describes the torus with $\tau$. This consideration fix the ambiguity. We also see (99) is modular invariant up to a phase, consistent with our discussion below (14), using the equations (14), (91) and (A.4).

Some examples for the local coordinates in the vertex region are shown in figure 7. The white region surrounded by the black curve is the local coordinate patch on the $z$-plane and its boundary is where the flat cylinder makes contact with the hyperbolic surface. Along with the local coordinate patch we have plotted the hyperbolic metric $e^\varphi$. Recall that its expressions is given by [40]

$$ds^2 = e^\varphi |dz|^2 = \lambda^2 \frac{|\partial w(z)|^2}{|w(z)|^2} \frac{|dz|^2}{\sin^2(\lambda \log |w(z)| + \lambda \log r)}. \tag{100}$$

Any choice of $w(z)$ in (100) leads to (possibly singular) hyperbolic metric. However, only $w(z)$ in (99) leads to the hyperbolic metric with geodesic boundary that is regular and singularity-free on the one-bordered torus. The regularity is equivalent to the double periodicity under the action of the lattice $\Lambda$ in this context. Indeed, figure 7 shows that using (99) gives an *almost* doubly-periodic metric. Since the problematic points are toward to the corners, it is reasonable to think that including higher-orders terms in the expansions achieves double periodicity. We observed increasing order of the expansion systematically improves this behavior.

## 6 Conclusion

In this paper, we have derived the local coordinates and vertex region of the hyperbolic tadpole vertex that is relevant for the one-loop diagrams in closed string field theory. To that end, we have considered a torus with a hyperbolic singularity which lead us to the Lamé equation (12). We have used the Polyakov conjecture (30) to fix its accessory parameters and use the formalism of [40] to obtain the relevant geometric data. We emphasize that only the *cubic*

information of CSFT, together with the input from the classical torus conformal blocks, was sufficient for our construction. As a byproduct, we uniformized the hyperbolic geometry on the one-bordered torus numerically and find the Weil-Petersson metric in the moduli space $\mathcal{M}_{1,1}(L)$ as an expansion in moduli. We have ran non-trivial checks and confirmed it leads to consistent results with the literature.

The hyperbolic tadpole vertex can be used to perform vacuum shift calculations in CSFT from first principles and this was our primary motivation behind this study. However, the vacuum shifts themselves are not observable [48] and there is no clear advantage of using hyperbolic vertices over other possible choices at this moment. The real advantage of using hyperbolic vertices is going to be apparent when the vacuum shift is considered in conjunction with the mass renormalization, which will yield physical results such as renormalized masses and S-matrix elements.[5] For example, the prime target here is the $SO(32)$ heterotic string theory on Calabi-Yau 3-folds [47], for which the perturbative vacuum shifts and the external states undergo mass renormalization.

However, the question of higher genus hyperbolic vertices remain open due to our poor understanding of classical conformal blocks for them. In particular, we need to solve for the hyperbolic geometry on the two-punctured torus in order to renormalize masses directly using the microscopic theory. Even though its Polyakov conjecture can be derived with relative ease (see appendix D), the classical conformal blocks necessary for the rest of the evaluation lacks. The most straightforward way to approach this problem would be to derive an (efficient) recursion relation for arbitrary blocks similar to the one given in [42] and take the semi-classical limit. Some steps towards it has been already undertaken in [75]. Despite these set-backs, we still see that the accessory parameter problem plays an important role in hyperbolic CSFT and a deeper theoretical understanding of them is certainly required.

Beyond motivations stemming from CSFT, it may be interesting to look at the Weil-Petersson Laplacian on the moduli space $\mathcal{M}_{1,1}(L)$, similar to what is done for $\mathcal{M}_{0,4}(0)$ in [45], as it is expected the spectrum to exhibit chaos. Another possibility for a future work is to investigate the results of [76,77] by evaluating the Laplacian on the surface *directly* and compare. Finally, it is possible to solve for the curvature of the Weil-Petersson metric over the moduli space using our techniques and this may be of interest in the theory of Riemann surfaces.

# Acknowledgments

I thank Harold Erbin and Barton Zwiebach for their insightful comments on the early draft and discussions. I also would like to thank Ted Erler, Åsmund Folkestad, Daniel Harlow, and Manki Kim for useful discussions.

**Funding information** This material is based upon work supported by the U.S. Department of Energy, Office of Science, Office of High Energy Physics of U.S. Department of Energy under grant Contract Number DE-SC0012567.

# A Special functions

Here we list the definitions, conventions, and properties of the special functions we have used throughout this work. For general references see [53, 78, 79]. The most important special

---

[5]These problems have been addressed using CSFT-inspired arguments in the past, see [72–74].

function for us was *the Weierstrass elliptic function* $\wp(z,\tau)$

$$\wp(z,\tau) \equiv \frac{1}{z^2} + \sum_{\lambda \in \Lambda \backslash \{0\}} \left[ \frac{1}{(z-\lambda)^2} - \frac{1}{\lambda^2} \right], \tag{A.1}$$

associated with the lattice $\Lambda = \mathbb{Z} + \tau \mathbb{Z}$ for $\tau \in \mathbb{H}$. By construction it satisfies

$$\wp(z,\tau) = \wp(z+1,\tau) = \wp(z+\tau,\tau), \qquad \wp(z,\tau) = \wp(z,\tau+1) = \frac{1}{\tau^2} \wp\left( \frac{z}{\tau}, -\frac{1}{\tau} \right). \tag{A.2}$$

This function has a Laurent expansion around $z = 0$ of the form

$$\wp(z,\tau) = \frac{1}{z^2} + \sum_{k=1}^{\infty} (2k+1) G_{2k+2}(\tau) z^{2k}, \quad \text{where} \quad G_{2k}(\tau) = \sum_{\lambda \in \Lambda \backslash \{0\}} \frac{1}{\lambda^{2k}}, \tag{A.3}$$

which converges for $|z| < 1$. Here $G_{2k}$ for $k \geq 1$ are called *the Eisenstein series*. They have the following modular transformations

$$G_{2k}(\tau+1) = G_{2k}(\tau), \qquad G_{2k}\left( -\frac{1}{\tau} \right) = \tau^{2k} G_{2k}(\tau), \tag{A.4}$$

and the *q*-expansions

$$G_{2k} = 2\zeta(2k) + \frac{4\zeta(2k)}{\zeta(1-2k)} \sum_{n=1}^{\infty} \sigma_{2k+1}(n) q^n, \quad \text{where} \quad \sigma_{2k+1}(n) = \sum_{d|n} d^{2k+1}. \tag{A.5}$$

Here $\zeta$ is the Riemann zeta function $\zeta(s) = \sum_{k=1}^{\infty} k^{-s}$ and $\sigma_{2k+1}(n)$ is the divisor sum function defined as the sum of the $(2k+1)$-th powers of the divisors of the integer $n$.

An auxillary function to the Weierstrass elliptic function is *the Weierstrass zeta function* $\zeta(z,\tau)$ (not to be confused with Riemann zeta function). This is defined as

$$\zeta(z,\tau) \equiv \frac{1}{z} + \sum_{\lambda \in \Lambda \backslash \{0\}} \left[ \frac{1}{z-\lambda} + \frac{1}{\lambda} + \frac{z}{\lambda^2} \right]. \tag{A.6}$$

It is clear $\wp(z,\tau) = -\partial_z \zeta(z,\tau)$ as given in (25a). This function is quasi-periodic, meaning

$$\zeta(z,\tau) = \zeta(z+1,\tau) - 2\zeta\left( \frac{1}{2} \right) = \zeta(z+\tau,\tau) - 2\zeta\left( \frac{\tau}{2} \right), \tag{A.7}$$

and it has the following modular transformations

$$\zeta(z,\tau) = \zeta(z,\tau+1) = \frac{1}{\tau} \zeta\left( \frac{z}{\tau}, -\frac{1}{\tau} \right). \tag{A.8}$$

These properties will be used in appendix D.

Furthermore, we encountered the Dedekind eta function, which is defined by

$$\eta(\tau) \equiv q^{\frac{1}{24}} \prod_{n=1}^{\infty} (1-q^n), \quad \text{where} \quad q = e^{2\pi i \tau}, \tag{A.9}$$

as usual. This immediately implies the identity (25c). The Dedekind eta function enjoys the following modular properties

$$\eta(\tau+1) = e^{i\pi/12} \eta(\tau), \qquad \eta\left( -\frac{1}{\tau} \right) = (-i\tau)^{1/2} \eta(\tau). \tag{A.10}$$

We have also made use of the odd Jacobi theta function

$$\vartheta_1(z|\tau) \equiv i \sum_{n=-\infty}^{\infty} (-1)^n q^{(n-1/2)^2/2} u^{n-1/2} = 2q^{\frac{1}{8}} \sin(\pi z) \prod_{n=1}^{\infty} (1-q^n)(1-uq^n)(1-u^{-1}q^n), \tag{A.11}$$

where $u = e^{2\pi i z}$. The functions $\zeta(z,\tau), \vartheta_1(z|\tau)$ and $\eta(\tau)$ can be related to each other as in (25a) or equivalently as [43, 56]

$$\wp(z,\tau) = -\partial_z^2 \log \vartheta_1(z|\tau) + 4\pi i \partial_\tau \log \eta(\tau). \tag{A.12}$$

Finally, we have used the polygamma functions in our evaluation of the saddle-point (39). These are defined as

$$\gamma^{(n)}(z) \equiv \frac{d^{n+1}}{dz^{n+1}} \log \Gamma(z), \quad \text{for} \quad n \geq 0. \tag{A.13}$$

In particular, their values at $z = 1, 1/2$ are evaluated in terms of the Riemann zeta function and the Euler–Mascheroni constant $\gamma$ as follows

$$\gamma^{(n)}(1) = \begin{cases} -\gamma, & \text{if} \quad n = 0, \\ (-1)^{n+1} n! \, \zeta(n+1), & \text{if} \quad n \geq 1, \end{cases} \tag{A.14a}$$

$$\gamma^{(n)}\left(\frac{1}{2}\right) = \begin{cases} -\gamma - 2\log 2, & \text{if} \quad n = 0, \\ (-1)^{n+1} n! \, (2^{n+1} - 1)\zeta(n+1), & \text{if} \quad n \geq 1. \end{cases} \tag{A.14b}$$

# B Classical torus conformal blocks

In this appendix we present our conventions and computation of the classical torus conformal blocks, based on the recursion relation derived in [42]. Begin with the torus conformal block,

$$\mathcal{F}_{c,\Delta'}^\Delta(q) = \frac{q^{\Delta' - \frac{c-1}{24}}}{\eta(\tau)} \left[ 1 + \sum_{n=1}^{\infty} q^n H_{1+6Q^2,\Delta'}^{\Delta,n} \right]. \tag{B.1}$$

The functions $\mathcal{F}_{c,\Delta'}^\Delta(q)$ are entirely determined by the Virasoro algebra and depends on the central charge $c$, as well as the conformal dimensions $\Delta, \Delta'$ of the external and internal operators. We are interested in their expansions in $q = e^{2\pi i \tau}$.

The coefficients $H_{1+6Q^2,\Delta'}^{\Delta,n}$ satisfy a recursion relation of the form [42]

$$H_{1+6Q^2,\Delta'}^{\Delta,n} = \sum_{1 \leq rs \leq n} \frac{A_{rs}(c)}{\Delta' - \Delta_{rs}(c)} P_{rs}(c,\Delta,\Delta_{rs} + rs) P_{rs}(c,\Delta,\Delta_{rs}) H_{c,\Delta_{rs}+rs}^{\Delta,n-rs}, \tag{B.2}$$

akin to the Zamolodchikov's recursion for the 4-point conformal blocks. Here the base case is given by $H_{1+6Q^2,\Delta'}^{\Delta,0} = 1$ and the ingredients for the recursion are given as follows. First, the function $A_{rs}(c)$ is defined as

$$A_{mn}(c) = \frac{1}{2} \left( \prod_{k=1-m}^{m} \prod_{l=1-n}^{n} \right)' \frac{1}{kb + \frac{l}{b}}, \tag{B.3a}$$

where the prime indicates that the terms with $(k,l) = (0,0), (m,n)$ are skipped in the product. Furthermore, we have

$$P_{mn}(c,\Delta_1,\Delta_2) = \prod_{\substack{p=1-m \\ p+m \text{ is odd}}}^{m-1} \prod_{\substack{q=1-n \\ p+n \text{ is odd}}}^{n-1} \frac{\alpha_1 + \alpha_2 + pb + \frac{q}{b}}{2} \frac{\alpha_1 - \alpha_2 + pb + \frac{q}{b}}{2}, \tag{B.3b}$$

with $\alpha_i$ is related to $\Delta_i$ through

$$\Delta_i = \frac{1}{4}(Q^2 - \alpha_i^2).$$ (B.3c)

Finally $\Delta_{rs}$ is defined by

$$\Delta_{rs}(c) = \frac{Q^2}{4} - \frac{1}{4}\left[rb + \frac{s}{b}\right]^2.$$ (B.3d)

The recursion relation (B.2) can be used to obtain the full torus conformal block (B.1). We don't report them since we are primarily interested in the classical torus conformal blocks $f_{\lambda'}^{\lambda}(q)$ given by the limit (37). We have computed $f_{\lambda'}^{\lambda}(q)$ to order $\mathcal{O}(q^4)$ for general $\lambda$ (and higher orders for the pre-determined values of $\lambda$). Its expressions is given by

$$
\begin{aligned}
f_{\lambda'}^{\lambda}(q) = {} & \frac{1}{4}\lambda'^2 \log q + \frac{(1+\lambda^2)^2}{8(1+\lambda'^2)}q \\
& + \frac{(1+\lambda^2)^2(-48(1+\lambda^2)(1+\lambda'^2)^2 + 96(1+\lambda'^2)^2(2+\lambda'^2) + (1+\lambda^2)^2(-7+5\lambda'^2)}{256(1+\lambda'^2)^3(4+\lambda'^2)}q^2 \\
& + \mathcal{O}(q^3).
\end{aligned}
$$ (B.4)

We refrain listing higher orders as the expressions get highly convoluted.

## C  Numerical details

We provide additional details on our numerical computations in this appendix. We begin with the investigation of the convergence of the series (53). Figure 8 shows the convergence of the length of the internal geodesic $2\pi\lambda_s$ computed using the double series expansion (53) for assorted values of $\tau$ and $\lambda$. We observe that the convergence is indeed achieved. It is interesting to take note that the perturbative series (that is, the series in $\xi$ (49)) plus the first few non-perturbative corrections (that is, the series in $\text{Re}(q)$) already generates the exact results in [59] to high accuracy, similar to [45]. This supports the validity of our procedure. Analogous convergence behavior has observed for the different values of $\tau$ and the remaining expansions in our study. We omit reporting them.

Next, we provide an evidence for the modular invariance of the accessory parameter $c$, see figure 9. Like in figure 2, the relative errors are relatively tiny and it is not really surprising given that we have also showed the modular invariance of $S_{HJ}^{(1,1)}$. Compared to the errors for the modular crossing of $S_{HJ}^{(1,1)}$ the errors here are about 3 orders of magnitude higher. This is expected, as we have taken a derivative of $S_{HJ}^{(1,1)}$ to obtain $c$. This has necessarily decreased the accuracy.

## D  The Polyakov conjecture for the $n$-bordered torus

In this appendix, we present the Polyakov conjecture for the torus with $n$ hyperbolic singularities for completeness. The relevant Fuchsian equation is given by

$$\partial^2\psi + \frac{1}{2}\left[\sum_{i=1}^n (\delta_i \wp(z-\xi_i,\tau) + \mu_i \zeta(z-\xi_i,\tau)) + c\right]\psi = 0,$$ (D.1)

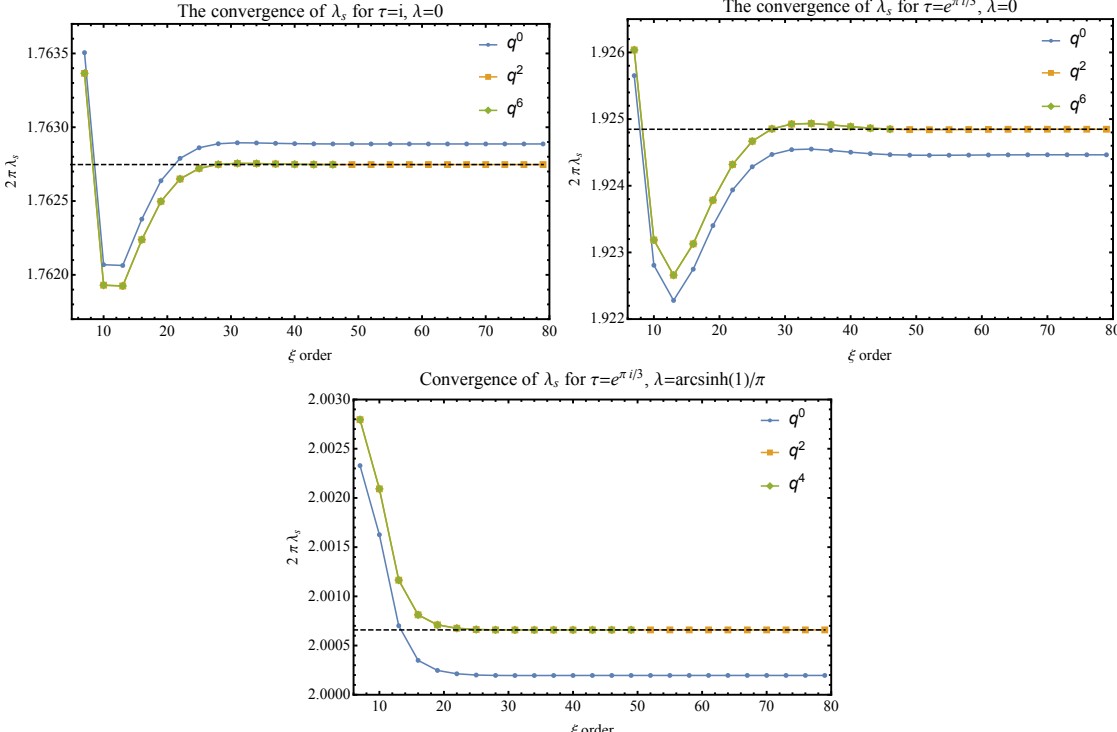

Figure 8: The progression of the convergence of the length of the internal geodesic with increasing orders in the expansion (53) . The dashed line indicates the exact result due to [59].

assuming the hyperbolic singularities are placed at $z = \xi_i$ and their associated classical weights are $\delta_i$ for $i = 1, \cdots n$. Here the reason for the appearance of the function $\wp(z - \xi_i, \tau)$ is exactly same as before, while the function $\zeta(z - \xi_i; \tau)$ appears as a consequence of having additional punctures and subsequent breaking of the $z \to -z$ symmetry when there was a single puncture placed at $z = 0$. This breaking allows us to include a simple pole at each puncture and its images, see (A.6). The inclusion of regular terms are still restricted by the double-periodicity. We point out this equation made an appearance in [80] in a different, but somewhat related, context.

The torus moduli $\tau$ and the position of the punctures $\xi_i$ are the complex moduli for fixed classical weights. There are $n$ independent moduli, upon noticing one of $\xi_i$ can be set to a fixed location by the translational invariance. Here $\tau$ is taking values in $\mathbb{H}/PSL(2, \mathbb{Z})$ as usual, while $\xi_i$ belongs to the set $(\mathbb{C} \setminus \{\xi_1, \cdots, \xi_{i-1}, \xi_{i+1} \cdots \xi_n\})/\Lambda$.

The variables $c$ and $\mu_i$ are the accessory parameters and they should be chosen so that real hyperbolic monodromies can be realized. There are $n$ of them given that they satisfy the constraint

$$\sum_{i=1}^{n} \mu_i = 0. \tag{D.2}$$

This can be argued by considering the contour integral of the expression inside the square bracket in (D.1) for which the contour surrounds all $n$ punctures. It is possible to deform this contour to contain all the images instead. The consistency then requires all residues to add up to zero, giving (D.2). When $n = 1$, $\mu_1 = 0$ as expected. Observe that this constraint guarantees the term inside the square bracket (D.1) is doubly-periodic, using the (quasi-)periodicity properties (A.2) and (A.7). This was the primary reason why we added $\zeta(z; \tau)$ function in (D.1) in order to include simple poles at the punctures and their images.

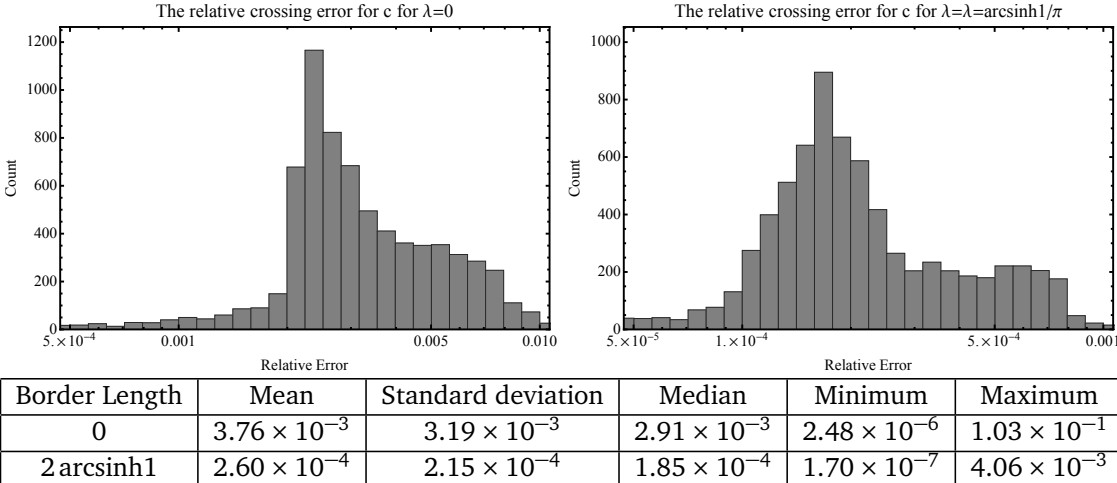

| Border Length | Mean | Standard deviation | Median | Minimum | Maximum |
|---|---|---|---|---|---|
| 0 | $3.76 \times 10^{-3}$ | $3.19 \times 10^{-3}$ | $2.91 \times 10^{-3}$ | $2.48 \times 10^{-6}$ | $1.03 \times 10^{-1}$ |
| $2\,\mathrm{arcsinh}1$ | $2.60 \times 10^{-4}$ | $2.15 \times 10^{-4}$ | $1.85 \times 10^{-4}$ | $1.70 \times 10^{-7}$ | $4.06 \times 10^{-3}$ |

Figure 9: The distribution of the relative errors for the crossing equation for the accessory (14) for $\lambda = 0$ and $\lambda = \mathrm{arcsinh}1/\pi$. The points are sampled from the fundamental domain with $\mathrm{Im}\,\tau < 1.2$.

We point out there is an involution symmetry of the form

$$\tau \to -\overline{\tau}, \quad \xi_i \to \overline{\xi_i} - \overline{\tau} \quad \implies \quad c \to \overline{c} + 2 \sum_{i=1}^{n} \mu_i \zeta\left(\frac{\overline{\tau}}{2}\right) = \overline{c}, \quad \mu_i \to \overline{\mu_i}, \tag{D.3}$$

after complex conjugating (D.1) and using (D.2). Note that shifting $\overline{\xi_i}$ was necessary in order to put $\xi_i$ back in the "fundamental region" of the torus after complex conjugation. This can be used to constrain the accessory parameters for sufficiently symmetric configurations.

Now consider the analog of the equation (26) when there are multiple insertions of $n$ hole operators $\mathcal{H}_{\lambda_i}$ with their associated conformal weights $\Delta_i$. This is given by

$$\left[\frac{1}{b^2}\partial_z^2 + (2\Delta_+ \eta_1(\tau) + 2\eta_1(\tau)z\partial_z) + \sum_{i=1}^{n} \big(\Delta_i \left(\wp\left(z - \xi_i, \tau\right) + 2\eta_1(\tau)\right) \tag{D.4}$$

$$+ \left(\zeta\left(z - \xi_i, \tau\right) + 2\eta_1(\tau)\xi_i\right)\partial_{\xi_i}\big) + 2\pi i \partial_\tau\right]\left\langle \phi_+(z)\prod_{i=1}^{n}\mathcal{H}_{\lambda_i}(\xi_i, \overline{\xi_i})\right\rangle_\tau + 2\pi i \partial_\tau \log Z(\tau) = 0.$$

In the semi-classical limit we expect the correlator to have a factorization of the form

$$\left\langle \phi_+(z)\prod_{i=1}^{N}\mathcal{H}_{\lambda_i}(\xi_i, \overline{\xi_i})\right\rangle_\tau \sim \phi_+^{cl}(z)\langle \Sigma_{1,n}\rangle_\tau \sim \phi_+^{cl}(z)\exp\left[-\frac{1}{2b^2}S_{HJ}^{(1,n)}(\tau, \xi_i)\right], \tag{D.5}$$

given $\phi_+(z)$ stays light while the hole operators get heavy. We have indicated the collection of hole operators simply by $\langle \Sigma_{1,n}\rangle_\tau$ above and evaluated it using the saddle-point approximation to the path integral. Here $S_{HJ}^{(1,n)}(\tau, \xi_i)$ is the (regularized) on-shell action relevant for this particular geometry. It depends on $\tau$ and the position of the punctures $\xi_i$ as well as their complex conjugates, but we haven't indicated the dependence on the latter for brevity.

Upon taking the semi-classical limit of (D.4) we get

$$\partial_z^2 \phi_+^{cl}(z) + \frac{1}{2}\bigg[\sum_{i=1}^{n}\big(\delta_i \wp\left(z - \xi_i, \tau\right) + 2\delta_i \eta_1(\tau) \tag{D.6}$$

$$- (\zeta(z - \xi_i, \tau) + 2\eta_1(\tau)\xi_i)\partial_{\xi_i}S_{HJ}^{(1,n)}(\tau, \xi_i)) - 2\pi i \partial_\tau S_{HJ}^{(1,n)}(\tau, \xi_i)\bigg]\phi_+^{cl}(z) = 0,$$

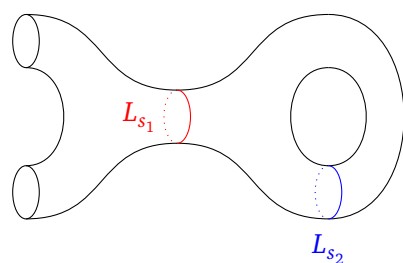

Figure 10: The two-bordered torus and its associated decomposition.

and the accessory parameters $c$ and $\mu_i$ are fixed by

$$c = 2\eta_1(\tau) \sum_{i=1}^{n} \Big[ \delta_i - \xi_i \, \partial_{\xi_i} S_{HJ}^{(1,n)}(\tau, \xi_i) \Big] - 2\pi i \partial_\tau S_{HJ}^{(1,n)}(\tau, \xi_i), \tag{D.7a}$$

$$\mu_i = -\partial_{\xi_i} S_{HJ}^{(1,n)}(\tau, \xi_i). \tag{D.7b}$$

This is the Polyakov conjecture for the torus with $n$ hyperbolic singularities. We point out the linear constraint (D.2) can be thought as consequence of the translational invariance due to the conformal Ward identity.

As a sanity check, let us test the modular invariance of the relation (D.7). The modular invariance of the equation (D.1) demands

$$T : c(\tau, \overline{\tau}) \to c(\tau + 1, \overline{\tau} + 1) = c(\tau, \overline{\tau}), \quad \mu_i(\tau, \overline{\tau}) \to \mu_i(\tau + 1, \overline{\tau} + 1) = \mu_i(\tau, \overline{\tau}), \tag{D.8a}$$

$$S : c(\tau, \overline{\tau}) \to c\left(-\frac{1}{\tau}, -\frac{1}{\overline{\tau}}\right) = \tau^2 c(\tau, \overline{\tau}), \quad \mu_i(\tau, \overline{\tau}) \to \mu_i\left(-\frac{1}{\tau}, -\frac{1}{\overline{\tau}}\right) = \tau \mu_i(\tau, \overline{\tau}). \tag{D.8b}$$

Furthermore, we have

$$T : \langle \Sigma_{1,1} \rangle_\tau \to \langle \Sigma_{1,1} \rangle_{\tau+1} = \langle \Sigma_{1,1} \rangle_\tau, \qquad S : \langle \Sigma_{1,1} \rangle_\tau \to \langle \Sigma_{1,1} \rangle_{-\frac{1}{\tau}} = |\tau|^{2 \sum_{i=1}^{n} \Delta_i} \langle \Sigma_{1,1} \rangle_\tau, \tag{D.9}$$

by the conformal weights of the hole operators, which subsequently produces

$$T : S_{HJ}^{(1,n)}(\tau, \xi_i) \to S_{HJ}^{(1,n)}(\tau + 1, \xi_i) = S_{HJ}^{(1,n)}(\tau, \xi_i), \tag{D.10a}$$

$$S : S_{HJ}^{(1,n)}(\tau, \xi_i) \to S_{HJ}^{(1,n)}\left(-\frac{1}{\tau}, \frac{\xi_i}{\tau}\right) = S_{HJ}^{(1,n)}(\tau, \xi_i) - 2 \sum_{i=1}^{n} \delta_i \log |\tau|. \tag{D.10b}$$

From these relations it is apparent that the Polyakov conjecture is indeed consistent with the modular invariance (D.7).

The relation (D.7) alone is not sufficient to construct the local coordinates for the $n$-punctured torus: the on-shell action $S_{HJ}^{(1,n)}$ has to be known as well. Constructing $S_{HJ}^{(1,n)}$ as function of the moduli requires the knowledge of the classical conformal blocks for the torus with arbitrary number of puncture. Assuming these blocks are obtained (for example, using a version of [75]) it is quite trivial to repeat an analogous bootstrap procedure to solve for $S_{HJ}^{(1,n)}$. For example, the two-bordered torus requires performing the decomposition shown in figure 10.

Lastly, let us make some further, more speculative, comments. Like in subsection 3.3, it is reasonable to *expect* that the on-shell action $S_{HJ}^{(1,n)}$ is the Kähler potential for the Weil-Petersson metric $g_{i\bar{j}}$ on the moduli space $M_{1,n}(L_i)$. More precisely, it is natural to conjure

$$g_{i\bar{j}} \sim \partial_i \partial_{\bar{j}} S_{HJ}^{(1,n)}, \tag{D.11}$$

where $i, \bar{j}$ stands for the moduli $\tau, \xi_i$ and their complex conjugates or their combinations thereof. Secondly, the ideas here may admit a generalization to the surfaces with genus greater than one. In this case the most natural way to present the Fuchsian equation would be on the hyperbolic upper half-plane $\mathbb{H}$, with demanding invariance under the Fuchsian group $\Gamma \subset PSL(2, \mathbb{R})$ associated with the surface $\Sigma_{g,n} \simeq \mathbb{H}/\Gamma$. During this construction the analogs of the functions $\wp(z, \tau)$ and $\zeta(z, \tau)$ for the Fuchsian groups will be needed and this will require introducing the sophisticated machinery of *automorphic forms* of the subgroups of $PSL(2, \mathbb{R})$, see [81]. It is quite probable that this may allow us to write down an appropriate Polyakov conjecture. Once this is done, we are again only bounded by our ability to compute the classical conformal blocks and the on-shell action as far as the expressions for the local coordinates and vertex regions are concerned.

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
