# Peer review of "Hyperbolic string tadpole"

_SciPost Physics, doi:SciPost Phys. 15, 237 (2023)_

## Round 1 · Referee Report · Matthew Headrick · 2023-11-15

Strengths
1. Makes important progress in closed string field theory.
2. Combines different methods in a novel way.
Weaknesses
N/A
Report
The once obscure subject of closed string string field theory has enjoyed a bit of a renaissance recently due to a few key advances. The paper under review builds on one of those, the proposal by Costello-Zwiebach to use hyperbolic metrics to define the string vertices. Specifically, this paper considers the simplest non-trivial quantum vertex, the once-punctured torus. Using tools from Liouville theory, combined with numerical work, the author uniformizes the metric and finds the local coordinates around the puncture and the Weil-Petterson metric on the moduli space.
This is an essentially technical advance, and the paper is written in a technical way, but it is an advance nonetheless. This work sets the stage for more complicated vertices, such as the twice-punctured torus, that have physical implications (e.g. mass renormalization). The paper is overall well organized and well written (aside from perhaps benefitting from a quick grammar check by a native speaker). I support its publication in SciPost.
Requested changes
Optionally, the paper could benefit from a quick grammar check.
Strengths
The paper is technically stong, involving nontrivial mathematics and provides explicit results.
Weaknesses
n.a.
Report
In this paper the author determines explicit data concerning the geometry of the hyperbolic tadpole vertex using the Polyakov conjecture to relate to classical Virasoro conformal blocks on the torus. The results are for the most part an application of ideas outlined in an earlier paper (2302.12843) giving a remarkable connection between hyperbolic vertices in closed SFT and Liouville theory.
There has been a fair amount of work on the off-shell closed string tadpole amplitude in the past. The discussion of 1704.01210 assumes the $SL(2,\mathbb{C})$ cubic vertex and is able to give a quite explicit description of the Feynman region of moduli space, but gives no natural definition of the tadpole vertex. The work of 1806.00450 determines the metric of minimal area on the 1-punctured torus at a special point in moduli space inside the tadpole vertex, but results are fully numerical and a complete characterization of the tadpole vertex is not given.
In this context the power of these techniques is truly impressive. The referee recommends the paper for publication.
Requested changes
n.a.

---

## Editorial Decision

published